# An investigation of irreproducibility in maximum likelihood phylogenetic inference

Xing-Xing Shen [1,2 ✉], Yuanning Li [3], Chris Todd Hittinger [4,5], Xue-xin Chen [1,2] & Antonis Rokas [3 ✉]

Phylogenetic trees are essential for studying biology, but their reproducibility under identical parameter settings remains unexplored. Here, we find that 3515 (18.11%) IQ-TREE-inferred and 1813 (9.34%) RAxML-NG-inferred maximum likelihood (ML) gene trees are topologically irreproducible when executing two replicates (Run1 and Run2) for each of 19,414 gene alignments in 15 animal, plant, and fungal phylogenomic datasets. Notably, coalescent-based ASTRAL species phylogenies inferred from Run1 and Run2 sets of individual gene trees are topologically irreproducible for 9/15 phylogenomic datasets, whereas concatenation-based phylogenies inferred twice from the same supermatrix are reproducible. Our simulations further show that irreproducible phylogenies are more likely to be incorrect than reproducible phylogenies. These results suggest that a considerable fraction of single-gene ML trees may be irreproducible. Increasing reproducibility in ML inference will benefit from providing analyses' log files, which contain typically reported parameters (e.g., program, substitution model, number of tree searches) but also typically unreported ones (e.g., random starting seed number, number of threads, processor type).

[1] State Key Laboratory of Rice Biology, Ministry of Agriculture Key Lab of Molecular Biology of Crop Pathogens and Insects, Zhejiang University, 310058 Hangzhou, China. [2] Institute of Insect Sciences, Zhejiang University, 310058 Hangzhou, China. [3] Department of Biological Sciences, Vanderbilt University, Nashville, TN 37235, USA. [4] Laboratory of Genetics, J. F. Crow Institute for the Study of Evolution, Wisconsin Energy Institute, Center for Genomic Science Innovation, University of Wisconsin-Madison, Madison, WI 53706, USA. [5] DOE Great Lakes Bioenergy Research Center, University of Wisconsin-Madison, Madison, WI 53706, USA. ✉email: xingxingshen@zju.edu.cn; antonis.rokas@vanderbilt.edu

The ability to replicate the results of a specific published experiment or analysis is a cornerstone of the scientific enterprise[1–3]. In the last few years, concerns about scientists' abilities to accurately reproduce the results of published studies in numerous disciplines, ranging from psychology and molecular biology to oncology, have steadily increased, leading to what some have dubbed as "the reproducibility crisis"[4–10].

Phylogenetics, the science of reconstructing evolutionary relationships of biological entities, is fundamental to the study of biology[11–14]. For example, phylogenetic trees are routinely used to understand how genes, genomes, organisms, and species evolve[15–17]. Application of phylogenetic trees also extends to other fields, and a phylogenetic framework has been employed to understand the evolution of diverse non-biological entities, such as languages and medieval manuscripts[18–21].

Concerns about reproducibility in phylogenetics are not new but have historically been attributed to the unavailability of the data used in inference[22,23]. For example, a 2013 meta-analysis reported that phylogenetic trees in 6277/7539 (83.3%) studies published in the last few decades are irreproducible due to the unavailability of the underlying data[22]. The availability of public data repositories (e.g., TreeBASE, Dryad, Figshare, Zenodo, OSF) coupled with the modernization of journal data sharing policies have greatly increased the availability of sequence alignment data, the resulting phylogenetic trees, as well as of information about program(s) and key parameter settings (e.g., substitution model) used[24–30].

Concerns remain, however, that the information that is now routinely provided in public data repositories may be still insufficient to ensure reproducibility of a study's findings. For example, phylogenetic studies seldom provide the random starting seed number used in inference, even though it is well established that it can affect hill-climbing tree heuristic searches of state-of-the-art, widely used maximum likelihood (ML)-based phylogenetic programs, such as IQ-TREE[31] and RAxML-NG[32]. More worrisomely, we currently know little about how the phylogenetic informativeness of the underlying data (e.g., the number of parsimony-informative sites or branch support values) or the variation in the computing resources used (e.g., number of the central processing unit (CPU) cores and type(s) of processor among studies or among nodes of a supercomputing cluster)[33,34] affect the reproducibility of phylogenetic inference.

Here we show that ~9 to ~18% of single-gene phylogenies are topologically irreproducible when executing two replicates (Run1 and Run2) on the same program (IQ-TREE or RAxML-NG on two threads of execution) with exactly the same parameter settings for each of 19,414 gene alignments in 15 animal, plant, and fungal studies. We further find that low phylogenetic informativeness (e.g., low percentage of parsimony-informative sites in gene alignment, short alignment length, and low branch support values), random starting seed number, processor type, and thread number contribute to the observed irreproducibility. Interestingly, thread number affected reproducibility in a program-specific manner.

## Results

**The ML phylogenies of a considerable number of genes in phylogenomic data sets are irreproducible.** To evaluate the reproducibility of single-gene phylogenetic trees, we collected 19,414 gene alignments from 15 animal, plant, and fungal phylogenomic data sets that span a wide spectrum of taxonomic ranks (Supplementary Data 1). For each gene alignment, we conducted two replicate runs (Run1 and Run2) using identical settings, including substitution model, random starting seed number, number of threads of execution (2), the number of

independent tree searches (20), log-likelihood epsilon value (0.0001), and ML program. Note that we chose values for the number of independent tree searches (20 instead of the standard range of 1–5) and log-likelihood epsilon (0.0001 instead of the standard 0.1) that aimed to minimize uncertainty in inference stemming from the fact that we are conducting heuristic searches. We used two state-of-the-art ML programs: IQ-TREE[31], whose heuristic search algorithm uses nearest-neighbor-interchange (NNI)[35], and RAxML-NG[32], whose algorithm uses subtree pruning and regrafting (SPR)[36] (Fig. 1).

For each pair of ML trees inferred from the Run1 and Run2 analyses of the same gene, we computed the normalized Robinson–Foulds[37] tree distance (nRFD; i.e., the fraction of

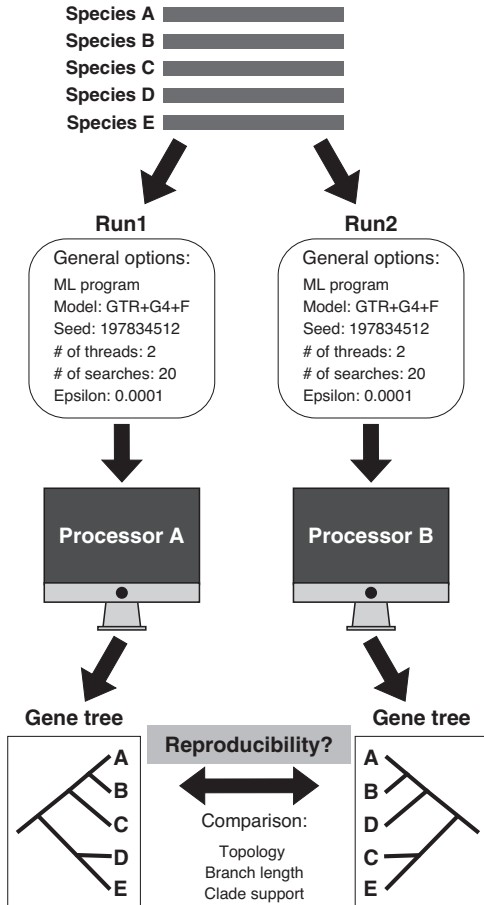

**Fig. 1 Overview of assessing the reproducibility of phylogenetic inference.** Our assessment begins with a gene sequence alignment. Two replicates (Run1 and Run2) using exactly the same parameters, including substitution model, random starting seed number, number of threads (2), number of tree searches (20), and log-likelihood epsilon for optimization (0.0001) on the same maximum likelihood (ML) program (IQ-TREE or RAxML-NG) were used to evaluate the reproducibility of the phylogenetic tree inferred from a given gene alignment. The Run1 and Run2 replicates were executed on two separate nodes (i.e., each analysis was run on a single node, but Run1 was executed on a different node than Run2) on a supercomputing cluster. Genes whose Run1 and Run2 generated topologically identical phylogenies were considered reproducible, while genes whose Run1 and Run2 generated topologically different phylogenies were considered irreproducible, but we also examined differences in the trees' branch lengths and clade support values. We analyzed 19,414 gene alignments from 15 animal, plant, and fungal phylogenomic data sets with a wide range of taxon sampling (from 15 to 1178 taxa) that were constructed using diverse gene sampling approaches.

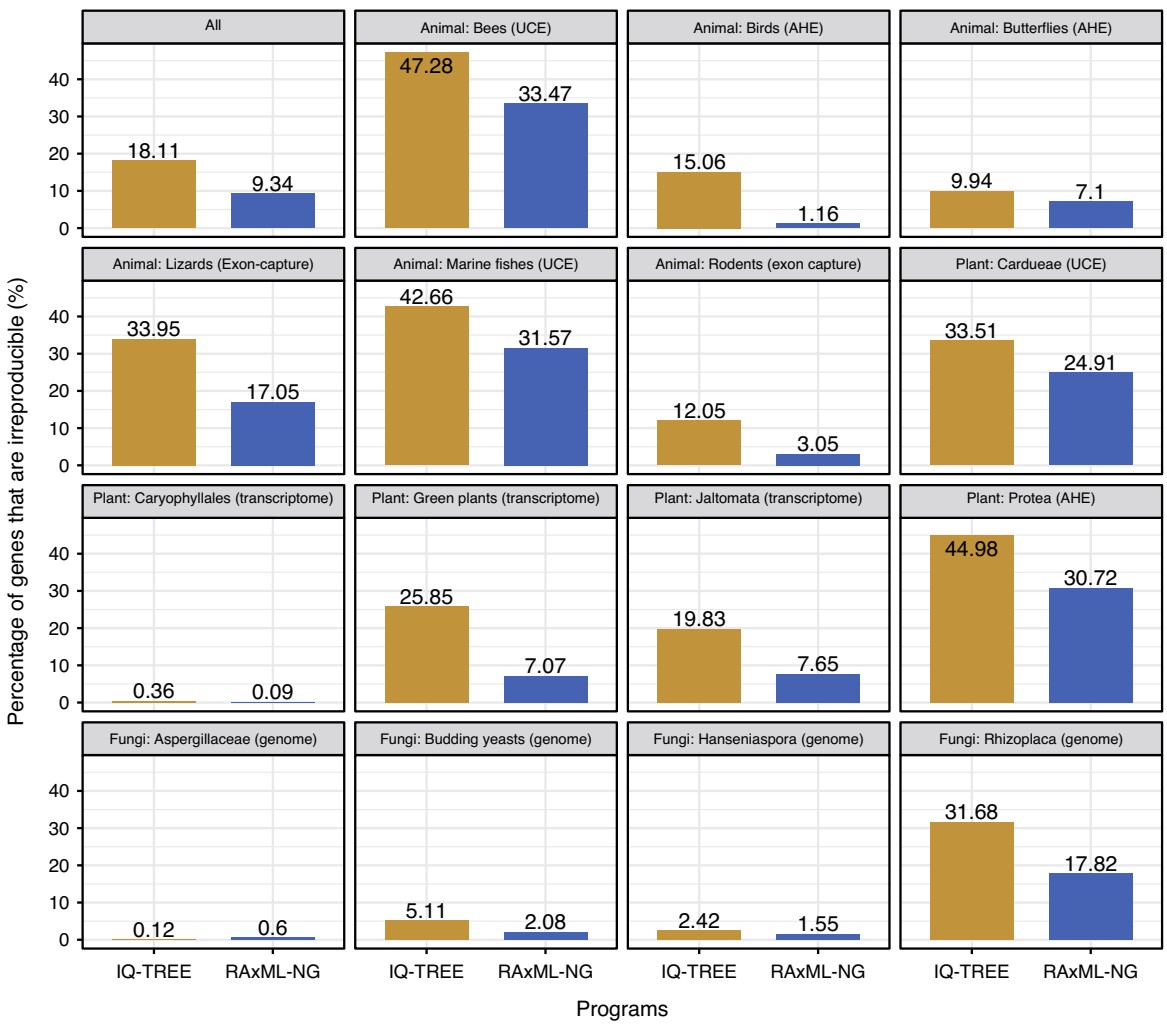

**Fig. 2 Considerable numbers of genes in phylogenomic data sets produce irreproducible phylogenies.** Bar plots show the percentages of genes that are irreproducible when using IQ-TREE (in yellow) and RAxML-NG (in blue), respectively. The bar plot at the upper left is based on all 19,414 gene alignments from 15 phylogenomic data sets; it shows that the phylogenies of 3515/19,414 genes (18.11%) and 1813/19,414 genes (9.34%) are irreproducible between two replicates (Run1 and Run2) using IQ-TREE (in yellow) and RAxML-NG (in blue), respectively. The rest of the bar plots show the individual results for each of the 15 phylogenomic data sets. These data sets were constructed using five different but widely accepted gene sampling approaches (shown in parentheses): Ultraconserved Element (UCE) capture, Anchored Hybrid Enriched (AHE) capture, conserved exon capture, transcriptome sequencing, and whole-genome sequencing. All 77,656 analyses (19,414 alignments * 2 replicates * 2ML programs) were run using two threads per node on the Center for High-Throughput Computing (CHTC) at the University of Wisconsin-Madison. Detailed values are given in Supplementary Data 2. All gene alignments, gene trees, and log files, as well as statistics of the results, are available on the figshare repository: https://doi.org/10.6084/m9.figshare.11917770.

bipartitions that differ between Run1 and Run2 trees) and the branch score distance of Kuhner and Felsenstein[38] (KF is the sum of the squares of the differences in branch lengths between Run1 and Run2 trees) with the R packages ape version 5.3 and phangorn version 2.5.5[39,40]. We found that 3515/19,414 genes (18.1%) in IQ-TREE and 1813/19,414 genes (9.3%) in RAxML-NG differed in their topologies (nRFD > 0) and branch lengths (KF > 0) (Figs. 1, 2 and Supplementary Fig. 1, Supplementary Data 1 and 2), 666/19,414 (3.4%) in IQ-TREE and 436/19,414 (2.2%) in RAxML-NG yielded the same topology (nRFD = 0) but differed in their branch lengths (KF > 0), while the remaining 15,233/19,414 (78.5%) in IQ-TREE and 17,165/19,414 (88.5%) in RAxML-NG had identical topology (nRFD = 0) and branch lengths (KF = 0) (Supplementary Fig. 1a). In addition, the differences in branch lengths between Run1 and Run2 trees with the same topology were much smaller than those observed between trees that differed in their topologies (Supplementary Fig. 1b). Observed patterns of topological irreproducibility were

very similar in two different supercomputing clusters at the University of Wisconsin-Madison and at Vanderbilt University (Fig. 2 and Supplementary Fig. 2, Supplementary Data 2 and 3).

Among the 15 phylogenomic data sets examined, the percentage of genes that produced topologically irreproducible phylogenetic trees varied between 0.12% and 47.28% (IQ-TREE) and between 0.09 and 33.47% (RAxML-NG) (Fig. 2). In 14/15 data sets, IQ-TREE had higher levels of gene tree irreproducibility than RAxML-NG (Fig. 2). To examine the degree to which the genes that produced topologically irreproducible trees inferred by the two programs overlapped, we compared IQ-TREE-inferred gene trees with RAxML-NG-inferred gene trees. We found that only 3940/19,414 (20.3%) gene alignments yielded topologically identical phylogenies in Run1 and Run2 of IQ-TREE and in Run1 and Run2 of RAxML-NG (Supplementary Fig. 3).

To examine whether the observed irreproducibility of gene trees resulted in significantly different topologies, we used the approximately unbiased (AU) test[41] to evaluate whether the

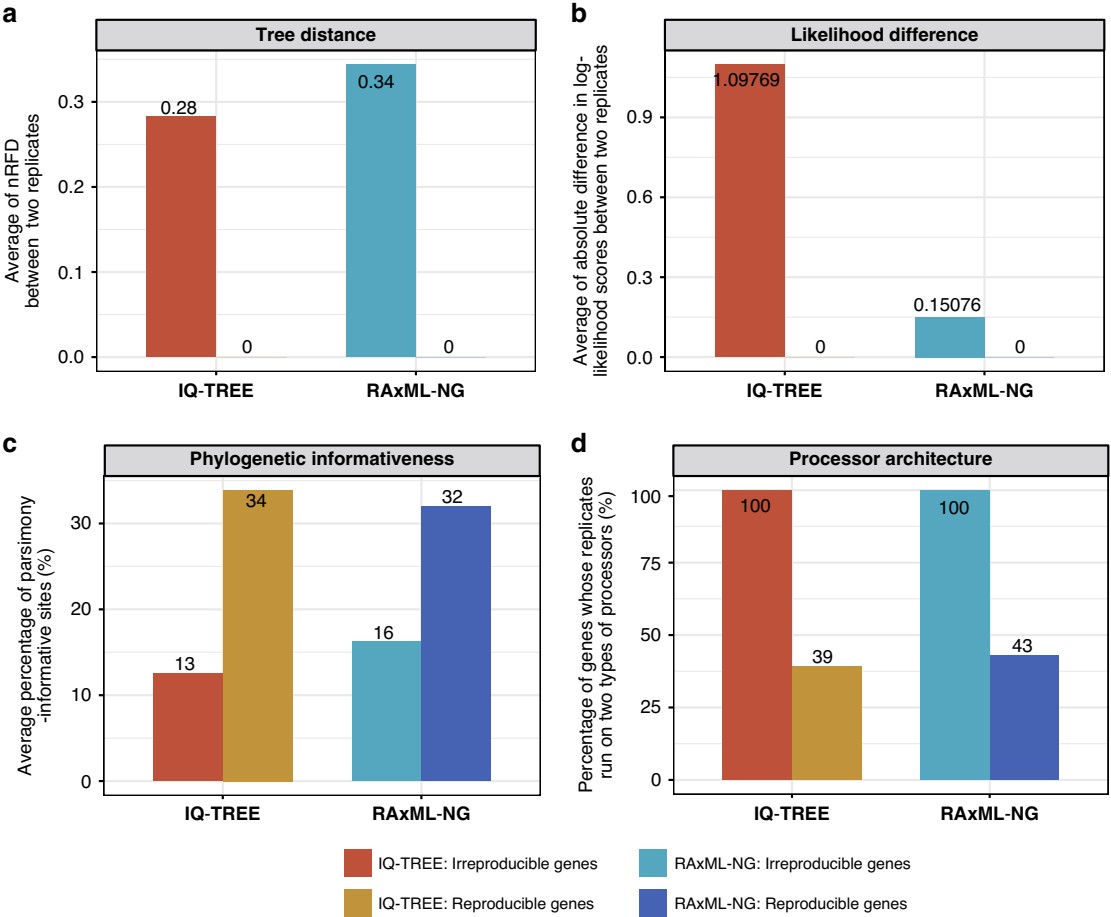

**Fig. 3 Key differences between genes that yield irreproducible phylogenies and those that yield reproducible ones.** Using IQ-TREE, 3515/19,414 (18.1%) gene alignments from 15 phylogenomic data sets yielded irreproducible phylogenies between two replicates (red bars), while the remaining 15,899 (81.9%) yielded reproducible phylogenies (yellow bars). Using RAxML-NG, 1813 (9.3%) genes yielded irreproducible phylogenies between two replicates (green bars), while 17,601 (90.7%) yielded reproducible phylogenies (blue bars). For the sets of reproducible and irreproducible genes in IQ-TREE and RAxML-NG analyses, **a** the normalized Robinson–Foulds tree distance (nRFD) between the gene trees inferred from the Run1 and Run2 replicates, **b** the absolute difference in log-likelihood values between the Run1 and Run2 replicates, **c** the percentage of a number of parsimony-informative sites in gene alignment, and **d** the percentage of Run1 and Run2 replicates executed on two separate 2-CPU nodes with different processor architectures (ML programs can automatically detect the kernel instructions on current processor architecture to best exploit the capabilities of CPU processor). **a**, **b** The degree to which the Run1 and Run2 trees of genes that yield irreproducible phylogenies differ in topology (**a**) and likelihood value (**b**). Irreproducible genes have a much lower average percentage of parsimony-informative sites (**c**) and run much more frequently on two different processors (**d**) than reproducible genes. Detailed values are given in Supplementary Data 2.

different topologies inferred in Run1 and Run2 analyses could equally explain the gene alignment (null hypothesis H0). We found that the Run1 and Run2 topologies for 302/3515 (8.59%) irreproducible single-gene ML phylogenies generated by IQ-TREE and 457/1813 (25.21%) irreproducible phylogenies generated by RAxML-NG were significantly different (AU test; $P$-value ≤ 0.05) (Supplementary Fig. 4). Among genes that yielded topologically irreproducible phylogenies, the percentage of those that reached significance in the AU test varied between 0.00 and 48% and between 0.00% and 59.26% across the 15 phylogenomic data sets examined in IQ-TREE and RAXML-NG (Supplementary Fig. 4), respectively.

Comparison of the Run1 and Run2 trees for the 3515 irreproducible gene phylogenies in IQ-TREE and for the 1813 irreproducible gene phylogenies in RAxML-NG (Fig. 3a, b and Supplementary Figs. 5–12) revealed considerable tree distance (average nRFD between Run1 and Run2 trees in IQ-TREE = 0.28; average nRFD in RAxML-NG = 0.34) (Fig. 3a and Supplementary Fig. 5) and log-likelihood score (average difference in log-

likelihood scores between Run1 and Run2 in IQ-TREE = 1.09769; the average difference in log-likelihood scores between Run1 and Run2 in RAxML-NG = 0.15076) differences (Fig. 3b and Supplementary Fig. 6).

Phylogenetic inference is a linear workflow that includes a series of separate steps (e.g., data sampling, orthology identification, multiple sequence alignment, tree-building)[42–44], with each step introducing uncertainty that can potentially affect the reproducibility of inference. For example, a recent study found that ML trees inferred by IQ-TREE and RAxML are more likely to be the optimal ones when more extensive tree searches are performed[45]. Examination of three representative phylogenomic data sets showed that the percentages of genes that produced reproducible phylogenies between two replicate runs when using 20 tree searches were highly similar to those when using 50 or 100 tree searches (Fig. 4), suggesting that the observed irreproducibility is over and above the uncertainty contributed by the heuristic nature of phylogenomic inference.

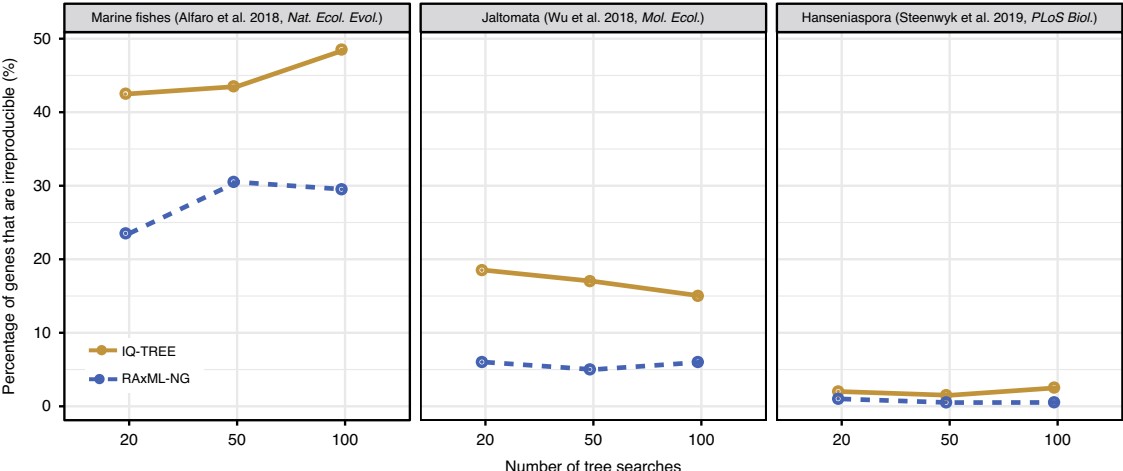

**Fig. 4 Effect of a number of tree searches on irreproducibility of single-gene trees.** Three representative studies in animals (marine fishes: 1001 genes and 120 taxa from Alfaro et al.[65]), plants (Jaltomata: 6431 genes and 15 taxa from Wu et al.[68]), and fungi (*Hanseniaspora*: 1033 genes and 29 taxa from Steenwyk et al.[69]) were used to examine the impact of different numbers of tree searches on the reproducibility of gene phylogenies. To reduce the computational burden of the analysis, we sampled the first 200 genes from each data set. The percentages of the 600 genes whose phylogenies are irreproducible when we ran two replicates (Run1 and Run2) on two separate nodes (i.e., each analysis was run on a 2-CPU node, but Run1 was executed on a different node than Run2) with an increasing number of tree searches on the CHTC cluster using IQ-TREE (in yellow) and RAxML-NG (in blue), respectively, are shown. These results suggest that increasing the number of tree searches used in our analyses has little effect on the irreproducibility of single-gene phylogenetic trees in both programs. All gene trees, log files, and statistics of the results, are available on the figshare repository: https://doi.org/10.6084/m9.figshare.11917770.

**Low phylogenetic informativeness, multithreading, and processor type contribute to irreproducibility.** To explore the underlying causes of the observed irreproducibility, we compared the characteristics of genes that yielded topologically irreproducible phylogenies (3515 in IQ-TREE and 1813 in RAxML-NG) to those that yielded topologically reproducible phylogenies (15,899 in IQ-TREE and 17,601 in RAxML-NG). We found that genes with lower phylogenetic informativeness (quantified by the percentage of parsimony-informative sites in gene alignment, alignment length, and branch support values) were substantially more likely to result in irreproducible phylogenies (Fig. 3c and Supplementary Figs. 7–9).

Specifically, genes whose Run1 and Run2 trees were topologically different had lower percentages of parsimony-informative sites (13% vs. 34% in IQ-TREE; 16% vs. 32% in RAxML-NG), shorter gene alignments (556 sites vs. 844 sites in IQ-TREE; 427 sites vs. 830 sites in RAxML-NG) and lower overall average bootstrap support values (58% vs. 71% in IQ-TREE; 25% vs. 50% in RAxML-NG), compared to genes whose Run1 and Run2 trees had the same topology (Fig. 3c and Supplementary Figs. 7, 8, and 9a). We further found that conflicting internal branches between the Run1 and Run2 trees of genes that yielded topologically irreproducible phylogenies received lower bootstrap support values (26% vs. 67% in IQ-TREE; 5% vs. 38% in RAxML-NG) than the congruent internal branches (Supplementary Fig. 9b). In contrast, the number of taxa in gene alignments had little impact on the irreproducibility of single-gene trees (Supplementary Fig. 10).

Examination of architectures of CPU processors revealed that genes run on two different types of processors (both programs can automatically detect the kernel instructions on current processor architecture to best exploit the capabilities of CPU processor) were more likely to result in topologically irreproducible phylogenies (Fig. 3d and Supplementary Fig. 11).

To test whether multithreading contributes to phylogenetic irreproducibility, we first assessed the reproducibility of single-gene phylogenies from 3819 genes from three large representative phylogenomic data sets on an increasing number of threads. We found that execution of Run1 and Run2 one right after the other on the same node yielded identical Run1 and Run2 tree topologies for all 3819 genes examined in both programs (Fig. 5a and Supplementary Data 4); we found the same result for both programs when using one or two thread(s) on the same node and for RAxML-NG when using up to five threads on the same node. In contrast, analyses using 3–5 threads on the same node with IQ-TREE yielded irreproducible single-gene trees for ~35.8% of 3819 genes (Fig. 5a, Supplementary Data 4). We obtained similar results when we executed Run1 and Run2 on a single laboratory server (Intel Xeon E5–2630 v3 @ 2.40 GHz processor with 16 threads) (Fig. 5b and Supplementary Data 4). These results suggest that multithreading using 3 or more threads contributes to irreproducibility in IQ-TREE but not in RAxML-NG.

The vast majority of phylogenomic analyses are performed across multiple nodes on supercomputing clusters that employ diverse processor types (rather than on the same node). To test the effect of multithreading on phylogenetic irreproducibility in this more realistic experimental design scenario, we executed Run1 and Run2 on two separate compute nodes (i.e., each analysis was run on a single node, but Run1 was executed on a different node than Run2) using an increasing number of threads. We found that the execution of Run1 and Run2 on two separate nodes resulted in an average irreproducibility of ~16.7% when using one or two thread(s) in IQ-TREE and RAxML-NG analyses (Fig. 5c). Interestingly, irreproducibility increased more than two-fold in IQ-TREE when using 3–5 threads (~35.8%), whereas irreproducibility in RAxML-NG analyses remained at similar levels (Fig. 5c and Supplementary Data 4). These results suggest that: (1) multithreading, coupled with the use of different nodes, is a major contributing factor to irreproducibility in IQ-TREE; and (2) the use of different nodes, but not multithreading, is a major contributing factor to irreproducibility in RAxML-NG.

**Irreproducibility of coalescent- and concatenation-based phylogenomic inference.** Since a considerable number of single genes yielded different Run1 and Run2 tree topologies, we examined the

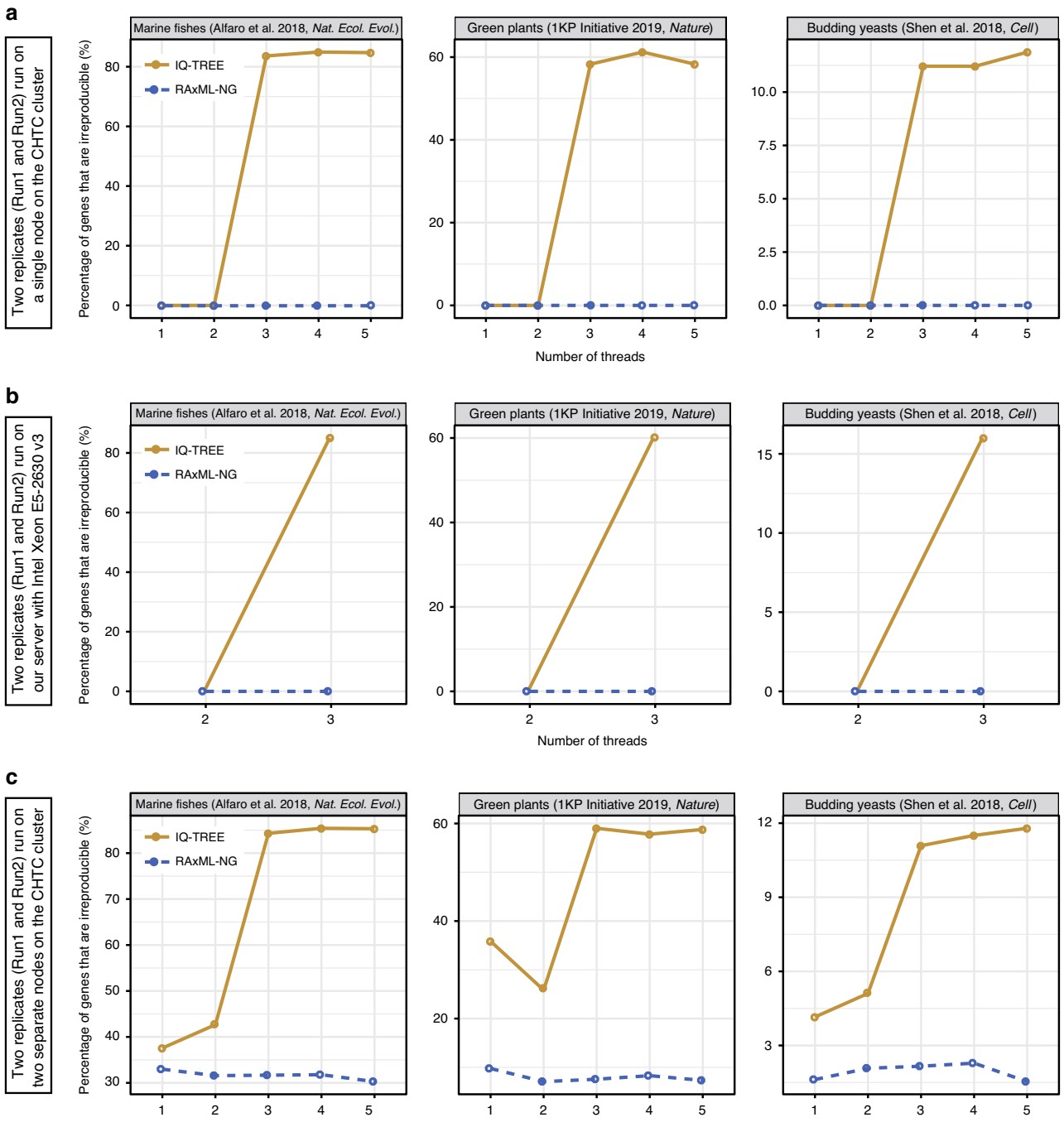

**Fig. 5 Variation in computing resources affects the irreproducibility of single-gene trees.** Three large representative studies in animals (marine fishes: 1001 genes and 120 taxa from Alfaro et al.[65]), plants (green plants: 410 genes and 1178 taxa from 1KP Initiative[66]), and fungi (budding yeasts: 2408 genes and 343 taxa from Shen et al.[16]) were used to examine the impact of threading and processor types on the reproducibility of gene phylogenies. Percentages of the 3819 genes whose phylogenies are irreproducible when **a** we ran two replicates (Run1 and Run2) on a single node (the two replicates were run one right after the other on the same node) on an increasing number of threads on the CHTC cluster using IQ-TREE (in yellow) and RAxML-NG (in blue), respectively. Since running all 3819 genes on a single laboratory server was computationally intractable, we sampled the first 200 genes from each data set. Percentages of these 600 genes when **b** we ran two replicates on a laboratory server (Intel Xeon E5–2630 v3 @ 2.40 GHz processor with 16 threads) on an increasing number of threads using IQ-TREE (in yellow) and RAxML-NG (in blue), respectively. Percentages of the 3819 genes whose phylogenies are irreproducible when **c** we ran two replicates (Run1 and Run2) on two separate nodes (i.e., each analysis was run on a single node, but Run1 was executed on a different node than Run2) on an increasing number of threads on the CHTC cluster using IQ-TREE (in yellow) and RAxML-NG (in blue), respectively. The irreproducibility of each gene was determined by comparing the topologies of single-gene trees generated by two replicates (Run1 and Run2). These results suggest that multithreading, coupled with the use of different nodes, is a major contributing factor to irreproducibility in IQ-TREE and that the use of different nodes, but not multithreading, is a major contributing factor to irreproducibility in RAxML-NG. Command lines and job submission files are given in Supplementary Note 1. All gene trees, log files, and statistics of the results, are available on the figshare repository: https://doi.org/10.6084/m9.figshare.11917770.

**Table 1 Comparisons of topologies, branch lengths, and internal branch support values between inferred phylogenies across two replicates.**

| Data set | Program | Coalescent-based ASTRAL species phylogenies (Run1 vs. Run2) | | | Concatenation-based ML species phylogenies (Run1 vs. Run2) | | |
|---|---|---|---|---|---|---|---|
| | | Tree distance (%)[a] | Branch distance[b] | Bipartitions with different support values (%)[c] | Tree distance (%)[a] | Branch distance[b] | Bipartitions with different support values (%)[c] |
| Animal: Bees | IQ-TREE | 22.5 | NA | NA | 0.0 | 1.46138E−07 | 2.7 |
| Animal: Birds | IQ-TREE | 1.5 | NA | NA | 0.0 | 4.09E−09 | 1.5 |
| Animal: Butterflies | IQ-TREE | 4.4 | NA | NA | 0.0 | 0.000283233 | 16.7 |
| Animal: Lizards | IQ-TREE | 3.8 | NA | NA | 0.0 | 1.49E−07 | 0.0 |
| Animal: Marine fishes | IQ-TREE | 6.8 | NA | NA | 0.0 | 1.03E−08 | 0.9 |
| Animal: Rodents | IQ-TREE | 0.0 | NA | 8.8 | 0.0 | 2.83E−10 | 0.0 |
| Plant: Cardueae | IQ-TREE | 4.9 | NA | NA | 0.0 | 2.45E−10 | 0.0 |
| Plant: Caryophyllales | IQ-TREE | 0.0 | NA | 0.0 | 0.0 | 0.0 | 0.0 |
| Plant: Green plants | IQ-TREE | 1.8 | NA | NA | 0.0 | 0.0 | NA |
| Plant: Jaltomata | IQ-TREE | 0.0 | NA | 0.0 | 0.0 | 8.00E−10 | 0.0 |
| Plant: Protea | IQ-TREE | 14.5 | NA | NA | 0.0 | 3.46E−10 | 0.0 |
| Fungi: Aspergillaceae | IQ-TREE | 0.0 | NA | 0.0 | 0.0 | 1.48E−07 | 0.0 |
| Fungi: Budding yeasts | IQ-TREE | 0.0 | NA | 1.8 | 0.0 | 0.0 | 0.0 |
| Fungi: Hanseniaspora | IQ-TREE | 0.0 | NA | 0.0 | 0.0 | 0.0 | 0.0 |
| Fungi: Rhizoplaca | IQ-TREE | 10.7 | NA | NA | 0.0 | 3.32E−10 | 0.0 |
| Animal: Bees | RAxML-NG | 8.6 | NA | NA | 0.0 | 0.0 | 0.0 |
| Animal: Birds | RAxML-NG | 1.5 | NA | NA | 0.0 | 0.0 | 0.0 |
| Animal: Butterflies | RAxML-NG | 2.0 | NA | NA | 0.0 | 0.0 | 0.0 |
| Animal: Lizards | RAxML-NG | 0.0 | NA | 30.8 | 0.0 | 0.0 | 0.0 |
| Animal: Marine fishes | RAxML-NG | 4.3 | NA | NA | 0.0 | 0.0 | 0.0 |
| Animal: Rodents | RAxML-NG | 0.0 | NA | 5.9 | 0.0 | 0.0 | 0.0 |
| Plant: Cardueae | RAxML-NG | 8.5 | NA | NA | 0.0 | 0.0 | 0.0 |
| Plant: Caryophyllales | RAxML-NG | 16.3 | NA | NA | 0.0 | 0.0 | 0.0 |
| Plant: Green plants | RAxML-NG | 0.7 | NA | NA | 0.0 | 0.0 | 0.0 |
| Plant: Jaltomata | RAxML-NG | 0.0 | NA | 8.3 | 0.0 | 0.0 | 0.0 |
| Plant: Protea | RAxML-NG | 21.0 | NA | NA | 0.0 | 0.0 | 0.0 |
| Fungi: Aspergillaceae | RAxML-NG | 0.0 | NA | 0.0 | 0.0 | 0.0 | 0.0 |
| Fungi: Budding yeasts | RAxML-NG | 0.0 | NA | 0.9 | 0.0 | 0.0 | NA |
| Fungi: Hanseniaspora | RAxML-NG | 0.0 | NA | 0.0 | 0.0 | 0.0 | 0.0 |
| Fungi: Rhizoplaca | RAxML-NG | 3.6 | NA | NA | 0.0 | 0.0 | 0.0 |

For each data set, coalescent-based ASTRAL trees were reconstructed from the Run1 and Run2 sets of individual gene trees; both Run1 and Run2 used identical settings, including substitution model, random starting seed number, number of independent tree searches, and ML program; concatenation-based ML trees were inferred twice from the supermatrix using IQ-TREE and RAxML-NG with identical settings.
[a]Percentage of bipartitions (or internal branches) that differ between two inferred species trees was quantified the normalized Robinson-Foulds tree distance.
[b]Branch distance between two inferred species trees was computed by the branch score distance of Kuhner and Felsenstein with the R packages ape and phangorn.
[c]The percentage of bipartitions (or internal branches) that received different bootstrap support values between two inferred species trees that were topologically identical (i.e., tree distance is 0). When topologies, branch lengths, or internal branch support values between two inferred phylogenies are identical, their estimated values are zero and are shown in bold. "NA" denotes "not applicable" due to either lack of external branch lengths in a coalescent-based ASTRAL tree, different topologies between two inferred species trees, or lack of standard bootstrap support values in the RAxML-NG analyses of the two largest data sets (plant: green plants and fungi: budding yeasts) due to computation cost.

**Table 2 Comparisons of coalescent-based ASTRAL species phylogenies with and without collapsing poorly supported bipartitions in replicate gene trees.**

| Data set | Program | Without collapsing branches | | With collapsing branches | |
|---|---|---|---|---|---|
| | | Conflicting bipartitions (%)[a] | Highly conflicting bipartitions (%)[b] | Conflicting bipartitions (%)[a] | Highly conflicting bipartitions (%)[b] |
| Animal: Bees | IQ-TREE | 22.46 | 1.87 | 18.18 | 1.60 |
| Animal: Birds | IQ-TREE | 6.09 | 0 | 6.60 | 0 |
| Animal: Butterflies | IQ-TREE | 7.35 | 0.49 | 5.39 | 0.74 |
| Animal: Lizards | IQ-TREE | 7.69 | 0 | 0 | 0 |
| Animal: Marine fishes | IQ-TREE | 7.69 | 0 | 13.68 | 0 |
| Animal: Rodents | IQ-TREE | 0 | 0 | 0 | 0 |
| Plant: Cardueae | IQ-TREE | 12.20 | 0 | 12.20 | 0 |
| Plant: Caryophyllales | IQ-TREE | 0 | 0 | 0 | 0 |
| Plant: Green plants | IQ-TREE | 5.70 | 0.38 | 5.53 | 0.55 |
| Plant: Jaltomata | IQ-TREE | 8.33 | 0 | 0 | 0 |
| Plant: Protea | IQ-TREE | 14.52 | 0 | 3.23 | 0 |
| Fungi: Aspergillaceae | IQ-TREE | 0 | 0 | 0 | 0 |
| Fungi: Budding yeasts | IQ-TREE | 0.29 | 0 | 0.29 | 0 |
| Fungi: Hanseniaspora | IQ-TREE | 0 | 0 | 0 | 0 |
| Fungi: Rhizoplaca | IQ-TREE | 3.57 | 0 | 0 | 0 |
| Animal: Bees | RAxML-NG | 27.27 | 1.87 | 8.02 | 0 |
| Animal: Birds | RAxML-NG | 4.06 | 0 | 0 | 0 |
| Animal: Butterflies | RAxML-NG | 6.86 | 0.49 | 6.86 | 0 |
| Animal: Lizards | RAxML-NG | 19.23 | 0 | 0 | 0 |
| Animal: Marine fishes | RAxML-NG | 3.42 | 0 | 5.98 | 0 |
| Animal: Rodents | RAxML-NG | 0 | 0 | 0 | 0 |
| Plant: Cardueae | RAxML-NG | 12.20 | 0 | 3.66 | 0 |
| Plant: Caryophyllales | RAxML-NG | 0 | 0 | 0 | 0 |
| Plant: Green plants | RAxML-NG | 0.26 | 0 | 1.02 | 0.09 |
| Plant: Jaltomata | RAxML-NG | 8.33 | 0 | 0 | 0 |
| Plant: Protea | RAxML-NG | 14.52 | 0 | 8.06 | 0 |
| Fungi: Aspergillaceae | RAxML-NG | 0 | 0 | 0 | 0 |
| Fungi: Budding yeasts | RAxML-NG | 0.29 | 0 | 0.59 | 0 |
| Fungi: Hanseniaspora | RAxML-NG | 0 | 0 | 0 | 0 |
| Fungi: Rhizoplaca | RAxML-NG | 3.57 | 0 | 0 | 0 |

Since running all 19,414 gene alignments from 15 phylogenomic data sets was computationally intractable, we sampled the first 100 genes from each data set. For each data set, coalescent-based ASTRAL trees were reconstructed from the Run1 and Run2 sets of 100 individual gene trees without and with collapsing branches with low bootstrap support values (≤50%); both Run1 and Run2 used identical settings, including substitution model, random starting seed number, number of threads of execution, number of independent tree searches, number of bootstrap replicates, and ML program.
[a]Percentage of conflicting bipartitions between coalescent-based ASTRAL species phylogenies inferred using Run1 and Run2 gene trees.
[b]Percentage of highly conflicting bipartitions (LPP ≥ 90%) between coalescent-based ASTRAL species phylogenies inferred using Run1 and Run2 gene trees.

coalescent-based ASTRAL species phylogenies reconstructed from the Run1 and Run2 sets of individual gene trees inferred using either IQ-TREE or RAxML-NG. We found that 9/15 (60%) phylogenomic data sets produced topologically different coalescent-based ASTRAL species phylogenies (Table 1 and Supplementary Fig. 12). When considering both topology and branch support values together, we found that 11/15 (73.3%) phylogenomic data sets whose gene trees were inferred using IQ-TREE and 13/15 (87%) data sets whose gene trees were inferred using RAxML-NG yielded different coalescent-based ASTRAL species phylogenies (i.e., nRFD > 0 and/or a number of internal branches with different support value > 0).

To examine the effects of removing poorly supported bipartitions from gene trees on the reproducibility of coalescent-based ASTRAL species phylogenies, we next collapsed branches with low bootstrap support values (≤50%) for Run1 and Run2 gene trees, and then used these partially multifurcating genes trees to infer coalescent-based ASTRAL species phylogenies. We found that collapsing branches with low bootstrap support values eliminated the irreproducibility of coalescent-based ASTRAL species phylogenies for three and four phylogenomic data sets when their gene trees were inferred by IQ-TEE and RAxML-NG, respectively, but eight phylogenomic data sets (IQ-TREE) and seven phylogenomic data sets (RAxML-NG) still

yielded topologically different ASTRAL species phylogenies (Table 2). These results suggest that the observed irreproducibility of coalescent-based ASTRAL species phylogenies remains even after we account for low phylogenetic informativeness.

We also examined whether concatenation-based ML trees that were inferred twice (Run1 and Run2) from the supermatrix using either IQ-TREE or RAxML-NG with identical settings were reproducible. We found that 15/15 phylogenomic data sets produced topologically identical concatenation-based ML species phylogenies (Table 1 and Supplementary Fig. 13). All phylogenomic data sets analyzed by RAxML-NG yielded phylogenies whose topologies, branch lengths, and branch support values were identical between Run1 and Run2 replicates. In contrast, only 4/15 (26%) phylogenomic data sets analyzed by IQ-TREE yielded phylogenies whose topologies, branch lengths, and branch support values were identical between Run1 and Run2 replicates. Whether coalescent-based phylogenetic inference or concatenation-based phylogenetic inference of species phylogenies is more reliable continues to be debated[46–48]. These results suggest that species phylogenies inferred using the coalescent-based approach, which relies on separately estimated gene trees, are more likely to be irreproducible than species phylogenies inferred using the concatenation-based approach, which relies on the supermatrix of gene alignments.

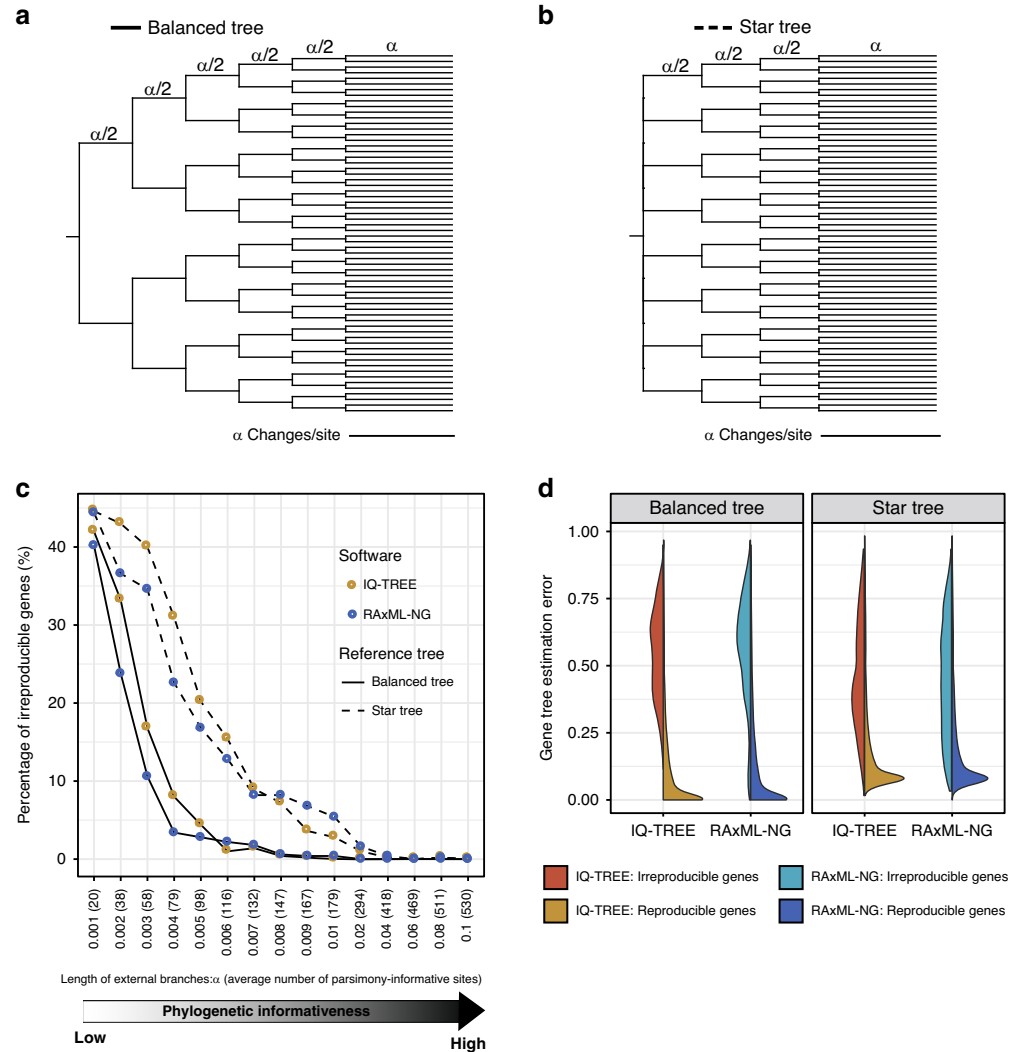

**Fig. 6 The topologies of genes yielding irreproducible phylogenies are more likely to be incorrect than those of genes yielding reproducible phylogenies.** To examine whether genes yielding irreproducible phylogenies were more likely to be incorrect than genes yielding reproducible phylogenies, we simulated gene sequence alignments on 30 phylogenetic trees (15 balanced trees and 15 star trees with 64 taxa) that were scaled by branch length $\alpha$ (where $\alpha = 0.001, 0.002, 0.003, 0.004, 0.005, 0.006, 0.007, 0.008, 0.009, 0.01, 0.02, 0.04, 0.06, 0.08,$ or $0.1$). Each phylogenetic tree was used to simulate 500 nucleotide sequence alignments, varying randomly in length from 300 to 1000 base pairs using Seq-Gen under the GTR model (-mGTR -a1 -g4 -i0 -f0.25,0.25,0.25,0.25 -n1 -or). For each of the 15,000 sequence alignments (30 phylogenetic trees * 500 sequence alignments), we assessed its reproducibility between two replicates (Run1 and Run2) on two separate nodes (i.e., each analysis was run on a single node, but Run1 was executed on a different node than Run2) using two threads on the CHTC cluster. **a** Balanced species tree of 64 taxa. Lengths of all external branches and internal branches are $\alpha$ and $\alpha/2$, respectively. **b** The star species tree of 64 taxa is a balanced species tree with six zero-length internal branches near the root. **c** Percentage of genes that yielded irreproducible phylogenies plotted against $\alpha$ value of reference tree used in the simulation. **d** Comparison of gene tree estimation error between genes yielding irreproducible phylogenies and genes yielding reproducible phylogenies. Gene tree estimation error corresponds to the normalized Robinson–Foulds tree distance (nRFD) between the inferred tree and the reference tree. Genes yielding irreproducible phylogenies are colored in red and genes yielding reproducible phylogenies are colored in yellow when using IQ-TREE; genes yielding irreproducible phylogenies are colored in green and genes yielding reproducible phylogenies are colored in blue when using RAxML-NG. This comparison shows the genes that generate irreproducible phylogenies are more likely to be incorrect than genes that generate reproducible phylogenies. We observed a similar magnitude of gene tree estimation error (even though irreproducibility for IQ-TREE was two-fold higher) when using three threads per node. The 15 balanced trees, the 15 star trees, command lines, gene alignments, and gene trees, as well as statistics of the results, are available on the figshare repository: https://doi.org/10.6084/m9.figshare.11917770.

**The genes that yield irreproducible phylogenies are more likely to be incorrect**. Since the true species and single-gene phylogenies for the 15 empirical phylogenomic data sets are unknown, it is impossible to assess whether irreproducible phylogenies are more or less similar to the true phylogeny than reproducible ones. To address this question, we simulated DNA sequence alignments on 15 balanced trees and 15 star trees, both with 64 taxa. Each tree was scaled by branch length $\alpha$ ($\alpha = 0.001, 0.002, 0.003, 0.004,$

$0.005, 0.006, 0.007, 0.008, 0.009, 0.01, 0.02, 0.04, 0.06, 0.08,$ or $0.1$), respectively; trees with larger $\alpha$ values had higher rates of substitutions per site and, consequently, higher average numbers of parsimony-informative sites (Fig. 6a, b). We then used each of these 15 balanced and 15 star trees to simulate 500 nucleotide sequence alignments that varied in length under the GTR + G4 model (detailed parameters are given in the file entitled "Seq_-Gen_run.bat" on the figshare repository: https://doi.org/10.6084/

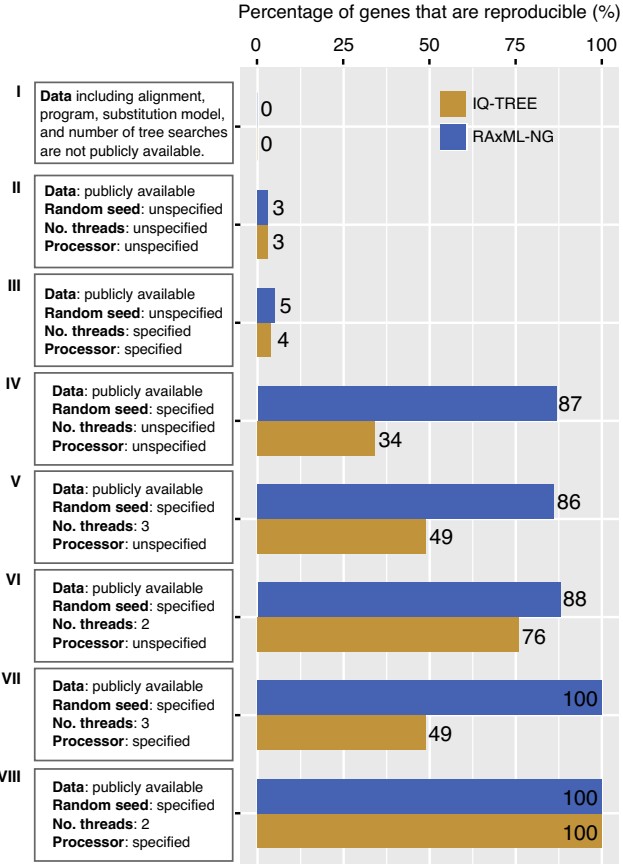

**Fig. 7 An overview of the reproducibility of phylogenetic inference.** The reproducibility of phylogenetic inference for eight specific scenarios: (I) The data and standard parameter settings typically reported in publications, including sequence alignment, program, substitution model, and a number of tree searches are not publicly available. (II) Sequence alignment, program, substitution model, and a number of tree searches are publicly available, but the number of threads, random starting seed number, and processor are not. (III) Sequence alignment, program, substitution model, number of tree searches, number of threads, and processor are publicly available, but random starting seed number is not. (IV) Sequence alignment, program, substitution model, number of tree searches, and random starting seed number are publicly available, but the number of threads and processors are not. (V) Sequence alignment, program, substitution model, number of tree searches, number of threads (3), and random starting seed number are publicly available, but the processor is not. (VI) Same scenario as V, but with two threads instead of three threads. (VII) Sequence alignment, program, substitution model, number of tree searches, number of threads (3), random starting seed number, and processor are publicly available. (VIII) Same scenario as VII, but with two threads instead of three threads. Analyses for each scenario utilized the first 200 genes from each of three large representative studies in animals (marine fishes: 1001 genes and 120 taxa from Alfaro et al.[65]), plants (green plants: 410 genes and 1178 taxa from 1KP Initiative[66]), and fungi (budding yeasts: 2408 genes and 343 taxa from Shen et al.[16]). Each gene's reproducibility of phylogenetic inference was assessed using two replicates (Run1 and Run2) for IQ-TREE (in yellow) and RAxML-NG (in blue), respectively. All analyses were performed on the CHTC cluster.

m9.figshare.11917770) using Seq-Gen.v1.3.2[49], generating a total of 15,000 DNA sequence alignments.

Analysis of these 15,000 simulated DNA sequence alignments showed that the percentage of genes yielding irreproducible phylogenies decreased with increasing α values (Fig. 6c and Supplementary Data 5), a result in agreement with the

observation from analysis of the 15 empirical phylogenomic data sets that genes producing irreproducible phylogenies tended to contain lower percentages of parsimony-informative sites (Fig. 3c). By comparing the 60,000 gene trees (15,000 gene alignments * 2 replicates * 2ML programs) against the reference tree used to simulate each gene alignment, we found that the 6104/60,000 (10%) irreproducible gene trees have three times higher average tree distance than the 53,896/60,000 (90%) reproducible gene trees (irreproducible gene phylogenies: average nRFD = 0.485 vs. reproducible gene phylogenies: average nRFD = 0.147) when using two threads per node (Fig. 6d and Supplementary Fig. 14). We observed a similar magnitude of gene tree estimation error (even though irreproducibility for IQ-TREE was two-fold higher) when using three threads per node (irreproducible gene phylogenies: average nRFD = 0.483 vs. reproducible gene phylogenies: average nRFD = 0.133) (Supplementary Data 5).

## Discussion
In this study, we found that ~9 to ~18% of single-gene phylogenies in 15 phylogenomic data sets were topologically irreproducible when analyzed by two widely used ML programs on a supercomputing cluster (Figs. 1 and 2). In addition to sequence alignment, program, substitution model, number of tree searches, and random starting seed number, which are known contributors, we further found that low phylogenetic informativeness, processor type, and multithreading contribute to the observed irreproducibility, with the effects of multithreading being program-specific.

Why did genes with low phylogenetic informativeness (e.g., a low percentage of parsimony-informative sites in gene alignment or lower average bootstrap support) more frequently fail to reproduce their topologies when running on different processor types (i.e., on processors with different kernel instructions) than highly informative genes (Fig. 3c, d and Supplementary Fig. 9a)? To achieve optimal performance, both IQ-TREE and RAxML-NG can automatically detect the kernel instruction on current processor architecture[50] to best exploit the capabilities of the CPU processor. Therefore, different processor architectures can result in otherwise identical replicates yielding different phylogenies with different log-likelihood values. While these differences in topology and likelihood are not generally sufficient to affect reproducibility in genes with high phylogenetic informativeness (Figs. 3c and 6c), our results suggest that they can have a dramatic effect on genes with low phylogenetic informativeness (Fig. 6d). Consistent with this explanation, irreproducibility was much higher (25.2%) in the eight data sets constructed using ultra-conserved element (UCE) capture[51], Anchored Hybrid Enriched (AHE) capture[52], or conserved exon capture[53] approaches, which tend to generate short alignments of highly conserved gene regions, than in the remaining seven studies (8.5%) (Supplementary Data 1 and 2). More broadly, data sets constructed from genes or regions of low phylogenetic informativeness and approaches biased toward including such regions may be particularly vulnerable to irreproducibility.

In contrast, phylogenetic trees of gene alignments executed on the same processor type (using one or two threads in IQ-TREE and any number in RAxML-NG) are reproducible (Fig. 5a, b). However, processor types vary both within, as well as between, supercomputing clusters. Thus, even if a study specified the processor type(s) used for inference, it may not always be straightforward (especially when new generations of processor types replace older ones) to reproduce results. Different containers (e.g., BIOS version, microarchitecture version, operating system, compiler manufacturer) could potentially influence the

compilation of ML program, further complicating reproducibility. Although irreproducibility can be eliminated in RAxML-NG by turning off the kernel auto-detection option (with the "–simd none" command; the current version of IQ-TREE does not offer this option), this results in substantially longer runtimes (average 2.4-fold increase in runtime; Supplementary Fig. 15).

Why did IQ-TREE have larger numbers of irreproducible phylogenies when using three or more threads, even when running on the same processor type (Fig. 5a, b)? This is likely because different orders of commutative addition of the per-site log-likelihoods when using three or more threads in IQ-TREE could result in two different phylogenies with different log-likelihood values. Consistent with this explanation, irreproducibility in RAxML-NG, which accounts for the difference of addition orders of the per-site log likelihoods[54,55], is not influenced by the use of three or more threads on the same processor (Fig. 5a, b).

Many previous studies have shown that most published gene alignments used to infer phylogenetic trees are inaccessible or have been permanently lost[22,24,26–30], effectively rendering these trees irreproducible (scenario I; Fig. 7). These studies have convincingly shown that the availability of gene alignments, phylogenetic trees, and key parameter settings (e.g., substitution model, number of tree searches, random starting seed number) can greatly improve the reproducibility of phylogenetic inference (scenario II; Fig. 7). The random starting seed number is a well-known contributor to uncertainty in tree inference in heuristic searches (scenarios III and IV; Fig. 7). Our study further shows that reproducibility of single-gene ML tree is also affected by processor type and multithreading, sometimes in a program-specific way (scenarios V–VIII; Fig. 7). Comparison of these scenarios allows assessment of the relative contribution of different parameters to irreproducibility. For example, a comparison of scenario III in Fig. 7, where all parameters except random starting seed number are identical between Run1 and Run2, against scenario IV, where all parameters are identical between the two replicates except a number of threads and processor, shows that random starting seed number differences lead to much lower reproducibility than differences in architecture and number of processors.

What is the impact of the single-gene tree and species tree irreproducibility that we observed in ML analyses in phylogenetic inference in general? The observed differences in the topologies of single-gene ML trees between replicate analyses were not trivial. At the level of single genes, we found that a considerable fraction of single-gene trees were different in their topologies between two runs with identical settings (Figs. 1 and 2), often significantly so (Supplementary Fig. 4), which could change the outcomes of analyses from phylogenetic programs that employ single-gene phylogenetic trees, such as ASTRAL[56,57], ModelTest-NG[58], Notung[59], and OrthoFinder[60].

At the level of phylogenomic inference, we found that the use of Run1 or Run2 gene trees was sufficient to change the ASTRAL species phylogeny inferred under a coalescent-based framework for several phylogenomic data sets (Tables 1 and 2). Importantly, collapsing of branches with low bootstrap support in individual gene trees reduced irreproducibility in species tree inference (Table 2). It is also important to emphasize that while IQ-TREE and RAxML-NG are widely used maximum likelihood programs, they are not the only ones. Whether other ML programs, such as FastTree[61], GARLI[62], MEGA[63], and PhyML[64], also exhibit irreproducibility, whether irreproducibility extends for other approaches, such as parsimony or Bayesian inference, and the impact of the observed irreproducibility in phylogenetic biology in general, are all topics that deserve further examination.

How can we increase the reproducibility of phylogenetic inference? One potential solution would be the mandatory reporting of not only sequence alignment (see also the recent commentary by Salomaki et al.[23]), program, substitution model, and number of independent tree searches, but also of random starting seed numbers, number of threads, and processor type used (Supplementary Note 2). However, the benefits need to be weighed against the practical difficulty of implementing this solution for the hundreds or thousands of gene alignments present in current phylogenomic data sets. Moving forward, a more practical alternative may be the releasing of the log file of each analysis, which contain a record of the values of all these key parameters (e.g., alignment, program name, number of tree searches, substitution model, type of processor, number of threads, and random starting seed).

## Methods

**Empirical phylogenomic data sets.** To assess the reproducibility of phylogenetic trees generated by individual genes, we retrieved 19,414 gene alignments from 15 phylogenomic studies in animals (6), plants (5), and fungi (4) (Supplementary Data 1). These 15 data sets were constructed using five widely accepted gene sampling approaches: Ultraconserved Element (UCE) capture, Anchored Hybrid Enriched (AHE) capture, conserved exon capture, transcriptome sequencing, and whole-genome sequencing. The 15 data sets comprise non-coding DNA (DNA), exon (DNA), and amino acid (AA) sequence alignments. The number of genes in these data sets ranges from 259 to 6431 with an average value of 1294; their number of taxa ranges from 15 to 1178 with an average value of 181. Finally, we note that this set of 15 data sets includes the largest available phylogenomic data sets in animals (Marine fishes: 1001 genes and 120 taxa from Alfaro et al.[65]), plants (Green plants: 410 genes and 1178 taxa from 1KP Initiative[66]), and fungi (budding yeasts: 2408 genes and 343 taxa from Shen et al.[16]). All gene alignments in FASTA form can be found on the figshare repository: https://doi.org/10.6084/m9.figshare.11917770.

**Assessment of reproducibility of single-gene phylogenetic trees.** Irreproducibility may manifest itself as a failure to reproduce published experiments (one's own experiments or ones performed by others). Assessing the reproducibility of published analyses in phylogenetics is extremely challenging because most published studies do not report the specific settings for key parameters (e.g., random starting seed number, number of threads of execution). Therefore, we focused on assessing whether we could reproduce our own results. Specifically, for each of 19,414 gene alignments, we utilized two replicates (Run1 and Run2) to assess whether the phylogenetic tree inferred by Run1 is topologically identical to the tree inferred by Run2 (Fig. 1). To take into account variation that may stem from different tree rearrangement algorithms used in heuristic searches, we used both the NNI-based IQ-TREE multi-thread version 1.6.8[31] and the SPR-based RAxML-NG multi-thread version 0.9.0[32] were used to infer a single-gene ML tree.

For nucleotide sequence alignments, we used the GTR + G4 + F substitution model; for amino acid sequence alignments, we used the best-fitting substitution model reported in the original study, which can be found on the figshare repository: https://doi.org/10.6084/m9.figshare.11917770. The two replicates used exactly the same parameter settings on the ML program used for inference, including gene alignment, program, substitution model, random starting seed number, number of threads (2), number of tree searches (20), and log-likelihood epsilon for optimization (0.0001).

Below are two specific examples of the specific command line instructions and parameter settings for the two replicates (Run1 and Run2) in IQ-TREE and RAxML-NG for a DNA sequence alignment (dna.fasta) and an amino acid alignment (aa.fasta):

DNA sequence alignment (dna.fasta):

iqtree -s dna.fasta -st DNA -m GTR + G4 + F -seed 4760742 -nt 2–runs 20 -me 0.0001 -pre iqtree_dna_Run1

iqtree -s dna.fasta -st DNA -m GTR + G4 + F -seed 4760742 -nt 2–runs 20 -me 0.0001 -pre iqtree_dna_Run2

raxml-ng–msa dna.fasta–search -msa-format FASTA–model GTR + G4 + F -seed 967956542–threads 2–tree pars{10},rand{10}–lh-epsilon 0.0001–prefix raxml-ng_dna_Run1

raxml-ng–msa dna.fasta–search -msa-format FASTA–model GTR + G4 + F -seed 967956542–threads 2–tree pars{10},rand{10}–lh-epsilon 0.0001–prefix raxml-ng_dna_Run2

Amino acid alignment (aa.fasta):

iqtree -s aa.fasta -st AA -m LG + G4 -seed 529418945 -nt 2–runs 20 -me 0.0001 -pre iqtree_aa_Run1

iqtree -s aa.fasta -st AA -m LG + G4 -seed 529418945 -nt 2–runs 20 -me 0.0001 -pre iqtree_aa_Run2

raxml-ng–msa aa.fasta–search -msa-format FASTA–model LG + G4 -seed 652954101–threads 2–tree pars{10},rand{10}–lh-epsilon 0.0001–prefix raxml-ng_aa_Run1

raxml-ng–msa aa.fasta–search -msa-format FASTA–model LG + G4 –seed 652954101–threads 2–tree pars{10},rand{10}–lh-epsilon 0.0001–prefix raxml-ng_aa_Run2

Overall, we executed 77,656 jobs (19,414 alignments * 2 replicates * 2 ML programs). Each job was run on a single node with 2 threads and 1 GB RAM on the Center for High-Throughput Computing (CHTC) at the University of Wisconsin-Madison (see command lines and job submission file in the Supplementary Note 1). In addition to the CHTC, we also ran these 77,656 jobs on the Advanced Computing Center for Research and Education (ACCRE) at Vanderbilt University.

All our tree topology comparisons used the normalized Robinson–Foulds[37] tree distance (nRFD), which we calculated using RAxML, version 8.2.3[67] (see command lines in Supplementary Note 1). For a given gene alignment, we considered that its tree was reproducible if the trees inferred from two replicates (Run1 and Run2) were topologically identical (i.e., nRFD = 0); in contrast, we considered the inferred tree irreproducible if the gene trees inferred from two replicates (Run1 and Run2) were topologically different from each other (i.e., nRFD > 0). Note that the reproducibility of each single-gene ML tree was only assessed by the same ML program (IQ-TREE or RAxML).

**Assessment of reproducibility of coalescent- and concatenation-based species trees**. For each of the 15 phylogenomic studies, we reconstructed their coalescent-based species trees from four sets of individual gene trees (iqtree_run1, iqtree_run2, raxml:ng_run1, and raxml:ng_run2) with ASTRAL version 5.6.3[56,57]. The reliability of each internal branch was evaluated using local posterior probability (LPP).

For a given phylogenomic study **Bee**, we obtained four sets of gene trees: Bee_iqtree_run1.genetrees, Bee_iqtree_run2.genetrees, Bee_raxml-ng_run1. genetrees, and Bee_raxml-ng_run2.genetrees. We used these four set to calculate their species trees using the following commands:
java -jar astral.jar -i Bee_iqtree_run1.genetrees -o Bee_iqtree_run1_lpp.tree
java -jar astral.jar -i Bee_iqtree_run2.genetrees -o Bee_iqtree_run2_lpp.tree
java -jar astral.jar -i Bee_raxml-ng_run1.genetrees -o Bee_raxml-ng_run1_lpp.tree
java -jar astral.jar -i Bee_raxml-ng_run2.genetrees -o Bee_raxml-ng_run2_lpp.tree

Concatenation-based ML trees were inferred twice (Run1 and Run2) from the supermatrix using either IQ-TREE or RAxML-NG with identical settings. The reliability of each internal branch was evaluated using 1000 ultrafast bootstrap replicates and 100 standard bootstrap replicates for IQ-TREE analysis and RAxML-NG, respectively. Below are two specific examples of the specific command line instructions and parameter settings for the two concatenation replicates (Run1 and Run2) in IQ-TREE and RAxML-NG for a DNA sequence alignment (dna.fasta) and an amino acid alignment (aa.fasta):

DNA sequence alignment (dna.fasta):
iqtree–runs 1 -nt 10 -st DNA -seed 369284957 -me 0.0001 -s dna.fasta -m GTR + G4 + F -bb 1000 -pre iqtree_dna_Run1
iqtree–runs 1 -nt 10 -st DNA -seed 369284957 -me 0.0001 -s dna.fasta -m GTR + G4 + F -bb 1000 -pre iqtree_dna_Run2
raxml-ng–force–all–threads 10–lh-epsilon 0.0001–seed 369284957–tree pars{1}–msa dna.fasta -msa-format FASTA–model GTR + G4 + F–bs-trees 100–prefix raxml-ng_dna_Run1
raxml-ng–force–all–threads 10–lh-epsilon 0.0001–seed 369284957–tree pars{1}–msa dna.fasta -msa-format FASTA–model GTR + G4 + F–bs-trees 100–prefix raxml-ng_dna_Run2

Amino acid alignment (aa.fasta):
iqtree–runs 1 -nt 10 -st AA -seed 369284957 -me 0.0001 -s aa.fasta -m LG + G4 -bb 1000 -pre iqtree_aa_Run1
iqtree–runs 1 -nt 10 -st AA -seed 369284957 -me 0.0001 -s aa.fasta -m LG + G4 -bb 1000 -pre iqtree_aa_Run2
raxml-ng–force–all–threads 10–lh-epsilon 0.0001–seed 369284957–tree pars{1}–msa aa.fasta -msa-format FASTA–model LG + G4–bs-trees 100–prefix raxml-ng_aa_Run1
raxml-ng–force–all–threads 10–lh-epsilon 0.0001–seed 369284957–tree pars{1}–msa aa.fasta -msa-format FASTA–model LG + G4–bs-trees 100–prefix raxml-ng_aa_Run2

Note that for the larger phylogenomic data sets (e.g., green plants: 410 genes and 1178 taxa from 1KP Initiative[66]; and budding yeasts: 2408 genes and 343 taxa from Shen et al.[16]), we used 32 threads instead of 10 threads (detailed parameters can be found in the log files on the figshare repository: https://doi.org/10.6084/m9.figshare.11917770).

**Impact of multithreading and processor types on gene tree reproducibility**. When the number of threads is not specified, IQ-TREE will automatically determine the best number of threads on the processor according to the length of the gene alignment and RAxML-NG will use all of the available cores on the processor. In addition, IQ-TREE and RAxML-NG can automatically detect the best kernel instruction on processor architecture to optimize the performance of the tree search, but different types of processor architectures can result in different kernel instructions. Therefore, we investigated whether increasing the number of threads and using different processor types affect the reproducibility of gene trees inferred by IQ-TREE and RAxML-NG. Because these analyses are computationally demanding, we performed them using a set of 3819 gene alignments from three large representative studies in

animals (marine fishes: 1001 genes[65]), plants (green plants: 410 genes[66]), fungi (budding yeasts: 2408 genes[16]). Specifically:

(i) For each of the 3819 gene alignments, two replicates (Run1 and Run2) were submitted to a single node (two replicates ran one right after the other on the same node) for 1, 2, 3, 4, and 5 thread(s), respectively. The total number of jobs executed on the CHTC cluster was 38,190 (3819 genes * 1 node (it contains 2 replicates) * 5 threading data points * 2ML programs) (see command lines and job submission file in Supplementary Note 1).

(ii) Since the analysis of phylogenomic data sets on a laboratory server is computationally intractable, we sampled the first 200 genes from each of the three data sets. For each of these 600 gene alignments, we ran two replicates (Run1 and Run2) on a laboratory server (Intel Xeon E5–2630 v3 @ 2.40 GHz processor with 16 threads) for 2 and 3 threads, respectively. All 3600 analyses (600 genes * 2 replicates * 2 threading data points * 2ML programs) were executed one right after the other on the server.

(iii) For each of the 3819 gene alignments, two replicates (Run1 and Run2) were submitted to two separate nodes (i.e., each analysis was run on a single node, but Run1 was executed on a different node than Run2) for 1, 2, 3, 4, and 5 thread(s), respectively. The total number of jobs executed on the CHTC cluster was 76,380 (3819 genes * 2 nodes (each contains 1 replicate) * 5 threading data points * 2ML programs) (see command lines and job submission file in Supplementary Note 1).

**Using simulated data to examine the accuracy of gene tree estimation for genes that yield irreproducible phylogenies**. To investigate the accuracy of gene tree estimation for genes whose phylogenies are irreproducible, we first generated 15 balanced trees and 15 star trees, both with 64 taxa, each of which was scaled by branch length $\alpha$ ($\alpha$ = 0.001, 0.002, 0.003, 0.004, 0.005, 0.006, 0.007, 0.008, 0.009, 0.01, 0.02, 0.04, 0.06, 0.08, or 0.1) (Fig. 6a, b), respectively. Next, each reference species tree of 64 taxa was used to generate 500 nucleotide gene alignments with varying length (randomized to be between 300 and 1000 base pairs) using Seq-Gen. v1.3.2[49] under the GTR + G4 model, shape for the gamma rate heterogeneity =1, proportion of invariable sites=0, and equal state frequency (-mGTR -a1 -g4 -i0 -f0.25,0.25,0.25,0.25 –l random length -n1 –z random seed) (detailed parameters are given in the file entitled "Seq_Gen_run.bat" on the figshare repository: https://doi.org/10.6084/m9.figshare.11917770).

For each simulated gene alignment, two replicates (Run1 and Run2) were executed on two separate nodes (i.e., each analysis was run on a single node, but Run1 was executed on a different node than Run2) using 2 threads on the CHTC cluster. Reproducibility of the resulting gene trees of the two replicates was assessed with RAxML, version 8.2.3. Furthermore, we calculated gene tree estimation error as the average of the nRFD values between each of the Run1 and Run2 gene trees and the reference species tree.

**Reporting summary**. Further information on research design is available in the Nature Research Reporting Summary linked to this article.

## Data availability
All gene alignments, gene trees, log files, and command lines, as well as summary and statistics of the runs, are available on the figshare repository: https://doi.org/10.6084/m9.figshare.11917770.

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

## Acknowledgements

We thank members of the Rokas lab and in particular Jacob Steenwyk, for helpful discussions and comments. This work was conducted in part using the resources of the Center for High-Throughput Computing (CHTC) at the University of Wisconsin-Madison and of the Advanced Computing Center for Research and Education (ACCRE) at Vanderbilt University. X.-X.S. was supported by the National Natural Science Foundation of China (No. 32071665) and the Fundamental Research Funds for the Central Universities (No. 2020QNA6019). C.T.H. was supported by the National Science Foundation (DEB-1442148), the USDA National Institute of Food and Agriculture (Hatch Project No. 1020204), in part by the DOE Great Lakes Bioenergy Research Center (DOE BER Office of Science No. DE-SC0018409), the Pew Charitable Trusts (Pew Scholar in the Biomedical Sciences), and the Office of the Vice-Chancellor for Research

and Graduate Education with funding from the Wisconsin Alumni Research Foundation (H. I. Romnes Faculty Fellow). X.-x.C. was supported by the Key International Joint Research Program of National Natural Science Foundation of China (No. 31920103005) and General Program of National Science Foundation of China (No. 31702035). A.R. was supported by the National Science Foundation (DEB-1442113), the National Institutes of Health/National Institute of Allergy and Infectious Diseases (1R56AI146096-01A1), the Guggenheim Foundation, and the Burroughs Wellcome Fund.

## Author contributions

Study conception and design: X.-X.S. and A.R.; acquisition of data: X.-X.S.; analysis and interpretation of data: X.-X.S., L.Y., C.T.H., X.-x.C., and A.R.; drafting of manuscript: X.-X.S. and A.R.; critical revision: X.-X.S., C.T.H., X.-x.C., and A.R.

## Competing interests

The authors declare no competing interests.
