## [Peer Review File · Nature Communications]

Reviewers' Comments:

Reviewer #1:

Remarks to the Author:

Review by Barbara Holland

This paper is the first to my knowledge that looks at irreproducibility in phylogenetics that occurs even when the software, data and model applied are identical. It will be of keen interest to many users of phylogenetic methods. I expect that it will also prompt further software development by the research teams that support Raxml, IQTree and similar methods.

The key claim of the paper, that phylogenetic methods are frequently irreproducible when run under (almost) identical conditions has been convincingly demonstrated and clearly presented.

My main recommendation in a revision would be to somewhat broaden the scope of the discussion of the results. There are a couple of extra analyses that would be useful to do in order to support this, but I don't want to suggest anything that will take months of computing time (the amount of computational effort that has gone into the paper is already considerable).

The results regarding species tree methods like ASTRAL are quite startling. It would be good to broaden the discussion around this point and to put the results of this paper in the context of the wider debate in the phylogenetic community around the relative merits of concatenation versus ILS aware summary methods.

See e.g.

David Bryant, Matthew W. Hahn. The Concatenation Question. Scornavacca, Celine; Delsuc, Frédéric; Galtier, Nicolas. Phylogenetics in the Genomic Era, No commercial publisher | Authors open access book, pp.3.4:1–3.4:23, 2020. fahal-02535651

To add in discussing this point it would be good to confirm that all 15 concatenated alignments were reproducible?

Also related to this issue, do edges that differ in run 1 and run 2 trees ever have high bootstrap support? Obviously it may be too computationally intensive to assess this for every irreproducible gene, but it might be interesting to pick a random sample of the genes that produced different trees for run 1 and run 2 and check the distribution of bootstrap support values for the bipartitions that differed.

Is a broader "moral of the story" that using methods that are reliant on accurately inferred bifurcating genes trees is fundamentally flawed?

Minor queries and typographical issues

Did you look for any differences in branch lengths between trees with the same topology? (panel b of fig 3 suggests that you did look and didn't find any differences). It is interesting that the irreproducibility issue never leads to finding different optima on a single tree.

If IQtree and Raxml both produce reproducible trees (i.e. their run 1 and run 2 are the same) do IQtree and Raxml always agree with each other?

If the processor and number of threads used were identical did this completely remove the issue of irreproducibility, or are there other factors at play as well? This is suggested but not explicitly stated.

Figure 3 panel d suggests that processor differences explain all of the instances of irreproducibility, how does this reconcile with the finding that threads matters for iqtree? Or does Figure 3 only relate to the runs with 2 threads? If so, it may be good to put this in the caption.

line 81

what is meant by "tree search number"? Is this the number of random starts?

line 92

what does it mean for two trees to be significantly different? I.e. what is the null hypothesis that is rejected by the AU test?

Line 108

what is normalised RF distance normalised by? Is a normalised RF of 0.58 saying that 58% of the splits differ?

line 241-244

I don't understand the explanation given here

"This is likely because different orders of addition of the per-site log likelihoods when using three or more threads in IQ-TREE could result in two different phylogenies with different log-likelihood values." addition is commutative so how can the result depend on the order?

line 310

searchers -> searches

line 370

what is a kernel vector?

line 593

produces -> produce

Reviewer #2:

Remarks to the Author:

The submission addresses the issue of reproducibility of phylogenetic tree inference. The authors consider several large published eukaryotic datasets. On each set of related genes, they infer two trees using the same program and parameters and check to which extent the topology between them agree. They observe differences in a substantial number of cases, and conclude that phylogenetic inference suffers from widespread irreproducibility, and conclude that authors should disclose random seed number, number of treads, and processor type used.

The work addresses an important problem, and the results obtained are thought-provoking.

I have one main concern, described below in detail, which I think could be satisfactorily addressed in a revision of the manuscript with more discussion and a restatement of the main conclusion.

Finally, note that I have reviewed an earlier version of the manuscript. I thank the authors for addressing some of the minor points I raised in my previous report. The points included in this report are, as far as I can tell, yet to be addressed.

Main point: what degree of reproducibility is desirable? and implications of the study

Phylogenetic methods are very broadly used, but the issue of interpreting trees, and in particular gauging the uncertainty around the inferred topology, is challenging even for experts (see e.g. intense debates around the placement of Ctenophora). Phylogenomic resources are complex pipelines, of which each step is afflicted by random and systematic errors.

The issue of reproducibility is thus highly relevant. But what exactly should be reproducible? In this paper, the authors have taken the view that reproducibility means to obtain the same tree topology every time. But, as I just mentioned, trees are subject to inference uncertainty, which is typically gauged using branch support measures such as bootstrap support. If the difference between two topologies is in uncertain parts of the tree (i.e. within the confidence interval, so to speak), is this still a reproducibility issue?

Because of the vast number of possible tree topologies (for a set of just 20 taxa, there are 221,643,095,476,699,771,875 possible unrooted binary topologies) and the complexity of optimising over the often rugged likelihood "landscape" of the topological space, it can be very difficult to find the global optimum. Inferred trees should never be considered at face value, but rather interpreted in light of the inference uncertainty.

Thus, unless the purpose is to debug code, I see relatively limited value in enforcing the ability to reproduce specific rounding-off errors or optimisation paths, as long as these differences don't materially affect the output trees *after accounting for inference uncertainty*.

There is, however, a real cost in demanding such narrow-sense reproducibility. As the authors themselves observed, avoiding variation due to CPU-specific optimisations mean that programs will typically run more slowly. More importantly, it would impose strong constraints on computing on clusters, as this would require keeping track of the number of jobs, order of computations, and CPU types employed for each subtask.

So while I think that it is interesting to demonstrate that multiple runs of the same data, using the same parameters, can result in different point estimates of the topology (which the authors convincingly show), I feel the conclusion should not be that all authors now need to systematically report their choice of seed and CPU types, and deactivate multithreading, but rather that more work is needed to assess whether these differences are significant in light of the uncertainty, and whether they might affect conclusions drawn from the data.

It might be possible to start answering this question from the results at hand. For instance, looking at the supplementary materials, I could see that some of the differences in the ASTRAL trees are on nodes with high branch support, which I think is noteworthy and cause for concern. See also the discretionary point below.

Minor points

* The result that RAxML-NG gives consistent results under multithreading but not in multiprocessing mode is intriguing. It may be worth inspecting the code or contacting the authors to find out why.

* Just to illustrate the practical difficulty of implementing the authors' own recommendation, I was unable to find the random seed values used to reconstruct the simulation in the "3_simulation_on_CHTC" supplementary file on FigShare. (I feel bad nitpicking this, given that I feel

the authors already went to considerable length to provide all supporting data, and this is a very minor oversight.)

* discretionary point: It would be interesting to quantify whether how that the difference in architectures and number of processors causes a difference larger than different random seeds

Christophe Dessimoz

Reviewer #3:

Remarks to the Author:

The paper looks at how reproduceable two phylogenetic analysis packages are, in terms of producing the same outcome when starting conditions are identical. It uses real and simulated data to demonstrate that the packages are not as repeatable as one would hope.

The programs the authors studied are primarily designed for building phylogenies quickly, the philosophy driving these programs is speed over accuracy, they are known to produce highly unreliable results, the program may find the highest likelihood less than 50% of the time for a given data set, as the authors of this study previously pointed out in 2018.

The title of the paper "Irreproducibility in molecular phylogenetics" is not justified given the analysis. The authors study two programs out of the hundreds available, the two they chose were known to be inaccurate. The authors then link this to the reproducibility crisis. They note that "... irreproducibility can be completely eliminated by turning off the kernel auto-detection option in RAxML-NG ..." but at the cost of a 2.4 speed penalty. If a crisis can be avoided by setting an option, it probably is not a crisis.

The authors suggest that 100% reproducibility can be obtained using recipe 10 in figures 6. It does not take into account a number of factors that will influence the results, especially if low level CPU instructions such as single instruction multiple data (SIMD) are part of the problem, to make the results reproducible, large amounts of technical information (which were not supplied) will be needed, BIOS versions, microarchitecture version, operating system / kernel version, compiler manufacturer and version, what version of the libraries the programs are using (this will be a long list with lots of dependencies), what implementation and version of MPI, compiler build options, etc. If reproducibility is the goal, technologies such as containers / VM's will be needed.

This paper is a technical note on the quirks of these two programs, not a look at reproducibility in molecular phylogenetics.

REVIEWER COMMENTS

Reviewer #1 (Remarks to the Author):

Review by Barbara Holland

This paper is the first to my knowledge that looks at irreproducibility in phylogenetics that occurs even when the software, data and model applied are identical. It will be of keen interest to many users of phylogenetic methods. I expect that it will also prompt further software development by the research teams that support Raxml, IQTree and similar methods.

The key claim of the paper, that phylogenetic methods are frequently irreproducible when run under (almost) identical conditions has been convincingly demonstrated and clearly presented.

AUTHORS' RESPONSE: We thank the reviewer for appreciating the importance of our work.

My main recommendation in a revision would be to somewhat broaden the scope of the discussion of the results. There are a couple of extra analyses that would be useful to do in order to support this, but I don't want to suggest anything that will take months of computing time (the amount of computational effort that has gone into the paper is already considerable).

The results regarding species tree methods like ASTRAL are quite startling. It would be good to broaden the discussion around this point and to put the results of this paper in the context of the wider debate in the phylogenetic community around the relative merits of concatenation versus ILS aware summary methods.

See e.g.

David Bryant, Matthew W. Hahn. The Concatenation Question. Scornavacca, Celine; Delsuc, Frédéric; Galtier, Nicolas. Phylogenetics in the Genomic Era, No commercial publisher |

Authors

open access book, pp.3.4:1–3.4:23, 2020. fihal-02535651

AUTHORS' RESPONSE: We thank the reviewer for the constructive suggestion. We added the concatenation-based maximum likelihood analyses for each dataset and included an additional section entitled "Irreproducibility of coalescent- and concatenation-based phylogenomic inference" that presents these results, cites some of the suggested studies, and discusses this important issue in the context of the debate between concatenation and coalescence.

As shown in the table below (Table 1 in the revised manuscript), we found that:

a) 9 / 15 (60%) phylogenomic datasets produced topologically different

coalescent-based ASTRAL species phylogenies when we inferred the Run1 and Run2 sets of individual gene trees with identical parameter settings in either IQ-TREE or RAxML-NG. When considering both topology and branch support values, we found that 11 / 15 (73.3%) phylogenomic datasets analyzed by IQ-TREE and 13 / 15 (87%) datasets analyzed by RAxML-NG yielded different coalescent-based ASTRAL species phylogenies.

- b) 15 / 15 phylogenomic datasets produced topologically identical concatenation-based ML species phylogenies when we inferred them twice independently from the same supermatrix using identical parameter settings in either IQ-TREE or RAxML-NG. All phylogenomic datasets analyzed by RAxML-NG yielded exactly the same concatenation-based ML species phylogeny, where the topology, branch length, and branch support are identical across Run1 and Run2. In contrast, only 4 / 15 (26%) phylogenomic datasets analyzed by IQ-TREE yielded exactly the same concatenation-based ML species phylogeny where the topology, branch length, and branch support are identical across Run1 and Run2.

Table 1 Comparisons of topologies, branch lengths, and internal branch support values between inferred phylogenies across two replicates

Dataset	Program	Coalescent-based ASTRAL species phylogenies			Concatenation-based ML species phylogenies		
		(Run1 vs Run2)			(Run1 vs Run2)		
		Tree distance (%) ^a	Branch distance ^b	Bipartitions with different support values (%) ^c	Tree distance (%) ^a	Branch distance ^b	Bipartitions with different support values (%) ^c
Animal: Bees	IQ-TREE	22.5	NA	NA	0.0	1.46138E-07	2.7
Animal: Birds	IQ-TREE	1.5	NA	NA	0.0	4.09E-09	1.5
Animal: Butterflies	IQ-TREE	4.4	NA	NA	0.0	0.000283233	16.7
Animal: Lizards	IQ-TREE	3.8	NA	NA	0.0	1.49E-07	0.0
Animal: Marine fishes	IQ-TREE	6.8	NA	NA	0.0	1.03E-08	0.9
Animal: Rodents	IQ-TREE	0.0	NA	8.8	0.0	2.83E-10	0.0
Plant: Cardueae	IQ-TREE	4.9	NA	NA	0.0	2.45E-10	0.0
Plant: Caryophyllales	IQ-TREE	0.0	NA	0.0	0.0	0.0	0.0
Plant: Green plants	IQ-TREE	1.8	NA	NA	0.0	0.0	NA
Plant: Jaltomata	IQ-TREE	0.0	NA	0.0	0.0	8.00E-10	0.0
Plant: Protea	IQ-TREE	14.5	NA	NA	0.0	3.46E-10	0.0
Fungi: Aspergillaceae	IQ-TREE	0.0	NA	0.0	0.0	1.48E-07	0.0
Fungi: Budding yeasts	IQ-TREE	0.0	NA	1.8	0.0	0.0	0.0

Fungi: Hanseniaspora	IQ-TREE	0.0	NA	0.0	0.0	0.0	0.0
Fungi: Rhizoplaca	IQ-TREE	10.7	NA	NA	0.0	3.32E-10	0.0
Animal: Bees	RAXML-NG	8.6	NA	NA	0.0	0.0	0.0
Animal: Birds	RAXML-NG	1.5	NA	NA	0.0	0.0	0.0
Animal: Butterflies	RAXML-NG	2.0	NA	NA	0.0	0.0	0.0
Animal: Lizards	RAXML-NG	0.0	NA	30.8	0.0	0.0	0.0
Animal: Marine fishes	RAXML-NG	4.3	NA	NA	0.0	0.0	0.0
Animal: Rodents	RAXML-NG	0.0	NA	5.9	0.0	0.0	0.0
Plant: Cardueae	RAXML-NG	8.5	NA	NA	0.0	0.0	0.0
Plant: Caryophyllales	RAXML-NG	16.3	NA	NA	0.0	0.0	0.0
Plant: Green plants	RAXML-NG	0.7	NA	NA	0.0	0.0	0.0
Plant: Jaltomata	RAXML-NG	0.0	NA	8.3	0.0	0.0	0.0
Plant: Protea	RAXML-NG	21.0	NA	NA	0.0	0.0	0.0
Fungi: Aspergillaceae	RAXML-NG	0.0	NA	0.0	0.0	0.0	0.0
Fungi: Budding yeasts	RAXML-NG	0.0	NA	0.9	0.0	0.0	NA
Fungi: Hanseniaspora	RAXML-NG	0.0	NA	0.0	0.0	0.0	0.0
Fungi: Rhizoplaca	RAXML-NG	3.6	NA	NA	0.0	0.0	0.0

For each dataset, coalescent-based ASTRAL trees were reconstructed from the Run1 and Run2 sets of individual gene trees; both Run1 and Run2 used identical settings, including substitution model, random starting seed number, number of threads of execution, number of independent tree searches, and ML program; Concatenation-based ML trees were inferred twice from the supermatrix using IQ-TREE and RAXML-NG with identical settings. ^a Percentage of bipartitions (or internal branches) that differ between two inferred species trees was quantified the normalized Robinson–Foulds tree distance. ^b Branch distance between two inferred species trees was computed by the branch score distance of Kuhner and Felsenstein with the R packages ape and phangorn. ^c The percentage of bipartitions (or internal branches) that received different bootstrap support values between two inferred species trees that were topologically identical (i.e., Topological difference is 0). When topologies, branch lengths, or internal branch support values between two inferred phylogenies are identical, their estimated values are zero and are shown in bold. “NA” denotes “not applicable” due to either lack of external branch lengths in coalescent-based ASTRAL tree, different topologies between two inferred species trees, or lack of standard bootstrap support values in the RAXML-NG analyses of the two largest datasets (Plant: Green plants and Fungi: Budding yeasts) due to expensive computation.

To add in discussing this point it would be good to confirm that all 15 concatenated alignments were reproducible?

AUTHORS’ RESPONSE: Our study’s focus was on reproducibility of the inferred

phylogenies and not on the reproducibility of the sequence alignments, which were retrieved from previous studies. All analyses used identical multiple sequence alignments so we can confirm that the observed irreproducibility does not stem from variation in the multiple sequence alignments used as input. However, the reviewer is correct that reproducibility of multiple sequence alignments is an additional potential variable that needs to be considered to achieve reproducible phylogenies. For example, a recent letter to the editor in *Nature Ecol. Evol.* (<https://www.nature.com/articles/s41559-020-01296-w>) stressed the importance of releasing uncurated sequence data. We have added a note of this issue in the last paragraph of the “Discussion” in the revised manuscript.

Also related to this issue, do edges that differ in run 1 and run 2 trees ever have high bootstrap support? Obviously it may be too computationally intensive to assess this for every irreproducible gene, but it might be interesting to pick a random sample of the genes that produced different trees for run 1 and run 2 and check the distribution of bootstrap support values for the bipartitions that differed.

AUTHORS’ RESPONSE: We thank the reviewer for the suggestion. To examine whether the bipartitions that differ between Run1 tree and Run2 tree were highly supported, we evaluated the support of each internal branch using 1,000 ultrafast bootstrap replicates (the option “-bb 1000”) and 100 standard bootstrap replicates (the option “-bs-trees 100”) for IQ-TREE analysis and RAxML-NG, respectively. Since running all 19,414 gene alignments from 15 phylogenomic datasets was computationally challenging, we sampled the first 100 genes from each dataset. For each gene alignment, we inferred gene trees twice with identical settings (see Supplementary Text).

As shown in the figure below (Supplementary Figure 9 in the revised manuscript), we found that:

- a) The genes that yield topologically irreproducible phylogenies have lower bootstrap support values than genes that yield topologically reproducible phylogenies (panel a in figure below).
- b) By comparing the branch support of conflicting internal branches against those of congruent internal branches within irreproducible genes, we found that conflicting internal branches received lower bootstrap support values than congruent internal branches (panel b in figure below).
- c) Furthermore, examination the frequency of internal branches, which were strongly supported by rapid bootstrap value of ≥ 95 in IQ-TREE analysis and standard bootstrap value of ≥ 70 in RAxML-NG analysis, showed that the percentage of conflicting internal branches with strong bootstrap support is much lower than that of congruent internal branches (panel c in figure below).

These new results are included in the second paragraph of the section entitled “Low phylogenetic informativeness, multithreading, and processor type contribute to irreproducibility” in the revised manuscript.

Is a broader "moral of the story" that using methods that are reliant on accurately inferred bifurcating genes trees is fundamentally flawed?

AUTHORS' RESPONSE: We thank the reviewer for raising this point. According to our results, we think that genes with low phylogenetic informativeness (e.g., low percentage of parsimony-informative sites in gene alignment or low average bootstrap support) are more likely to be irreproducible in their topology. Moreover, compared to reproducible bipartitions, irreproducible bipartitions tend to have lower bootstrap support. Using inferred bifurcating gene trees from Run1 and Run2 to infer coalescent-based ASTRAL species phylogenies led to irreproducibility in 9 / 15 datasets (Table 1).

To examine whether irreproducibility of coalescent-based ASTRAL species phylogenies could be reduced by removing poorly supported bipartitions from the Run1 and Run2 trees of single genes, we collapsed branches with low bootstrap support values ($\leq 50\%$) for Run1 and Run2 gene trees, and then used these partially multifurcating genes trees to infer coalescent-based ASTRAL trees. As shown in the Table below (Table 2 in the revised manuscript), we found that the percentage of conflicting bipartitions across coalescent-based ASTRAL species phylogenies from the Run1 and Run2 sets of individual gene trees, ranges from 0.29% to 18.18%. Moreover, collapsing branches with low bootstrap support values eliminated the irreproducibility of coalescent-based ASTRAL species phylogenies for three and four phylogenomic datasets in IQ-TEE and in RAxML-NG, respectively. However, eight phylogenomic datasets in IQ-TREE and seven phylogenomic datasets in RAxML-NG still yielded topologically different ASTRAL species phylogenies.

These results suggest that the observed irreproducibility is sufficiently strong to alter the output trees even if we remove poorly supported branches from our gene trees. However, they also suggest that reducing the impact of lowly supported branches in genes trees can eliminate species tree irreproducibility in some datasets. We have included these new results in the second paragraph of the section “Irreproducibility of coalescent- and concatenation-based phylogenomic inference” in the revised manuscript.

Table 2 Comparisons of coalescent-based ASTRAL species phylogenies with and without collapsing poorly supported bipartitions in replicate gene trees.

Dataset	Program	Without collapsing branches		With collapsing branches	
		Conflicting bipartitions (%) ^a	Highly conflicting bipartitions (%) ^b	Conflicting bipartitions (%) ^a	Highly conflicting bipartitions (%)
Animal: Bees	IQ-TREE	22.46	1.87	18.18	1.60
Animal: Birds	IQ-TREE	6.09	0	6.60	0
Animal: Butterflies	IQ-TREE	7.35	0.49	5.39	0.74
Animal: Lizards	IQ-TREE	7.69	0	0	0
Animal: Marine fishes	IQ-TREE	7.69	0	13.68	0
Animal: Rodents	IQ-TREE	0	0	0	0
Plant: Cardueae	IQ-TREE	12.20	0	12.20	0
Plant: Caryophyllales	IQ-TREE	0	0	0	0
Plant: Green plants	IQ-TREE	5.70	0.38	5.53	0.55
Plant: Jaltomata	IQ-TREE	8.33	0	0	0
Plant: Protea	IQ-TREE	14.52	0	3.23	0
Fungi: Aspergillaceae	IQ-TREE	0	0	0	0
Fungi: Budding yeasts	IQ-TREE	0.29	0	0.29	0
Fungi: Hanseniaspora	IQ-TREE	0	0	0	0

Fungi: Rhizoplaca	IQ-TREE	3.57	0	0	0
Animal: Bees	RAxML-NG	27.27	1.87	8.02	0
Animal: Birds	RAxML-NG	4.06	0	0	0
Animal: Butterflies	RAxML-NG	6.86	0.49	6.86	0
Animal: Lizards	RAxML-NG	19.23	0	0	0
Animal: Marine fishes	RAxML-NG	3.42	0	5.98	0
Animal: Rodents	RAxML-NG	0	0	0	0
Plant: Cardueae	RAxML-NG	12.20	0	3.66	0
Plant: Caryophyllales	RAxML-NG	0	0	0	0
Plant: Green plants	RAxML-NG	0.26	0	1.02	0.09
Plant: Jaltomata	RAxML-NG	8.33	0	0	0
Plant: Protea	RAxML-NG	14.52	0	8.06	0
Fungi: Aspergillaceae	RAxML-NG	0	0	0	0
Fungi: Budding yeasts	RAxML-NG	0.29	0	0.59	0
Fungi: Hanseniaspora	RAxML-NG	0	0	0	0
Fungi: Rhizoplaca	RAxML-NG	3.57	0	0	0

For each dataset, coalescent-based ASTRAL trees were reconstructed from the Run1 and Run2 sets of individual gene trees without and with collapsing branches with low bootstrap support values ($\leq 50\%$); both Run1 and Run2 used identical settings, including substitution model, random starting seed number, number of threads of execution, number of independent tree searches, number of bootstrap replicates, and ML program. ^a Percentage of conflicting bipartitions between coalescent-based ASTRAL species phylogenies inferred using Run1 and Run2 gene trees. ^b Percentage of highly conflicting bipartitions ($LPP \geq 90\%$) between coalescent-based ASTRAL species phylogenies inferred using Run1 and Run2 gene trees.

Minor queries and typographical issues

Did you look for any differences in branch lengths between trees with the same topology? (panel b of fig 3 suggests that you did look and didn't find any differences). It is interesting that the irreproducibility issue never leads to finding different optima on a single tree.

AUTHORS' RESPONSE: We thank the reviewer for the suggestion. To check whether branch lengths between Run1 and Run2 trees with the same topology are same, we computed the normalized Robinson–Foulds tree distance (nRFD) and branch score distance of Kuhner and Felsenstein (KF) between Run1 and Run2 trees with the R packages ape and phangorn. In IQ-TREE analysis, we found that 3,515 / 19,414 (18.1%) gene alignments from 15 phylogenomic datasets yielded different topologies (nRFD > 0) and different branch lengths (KF > 0) between two replicates (shown in red in the figure below), 666 / 19,414 (3.4%) yielded the same topology (nRFD = 0) and different branch lengths (KF > 0) (yellow), while the remaining 15,233 / 19,414 (78.5%) yielded the same topology (nRFD = 0) and identical branch lengths (KF = 0) (green). In RAxML-NG analysis, we found that 1,813 / 19,414 (9.3%) genes yielded different topologies (nRFD > 0) and different branch lengths (KF > 0) between two replicates (red), 436 / 19,414 (2.2%) yielded the same topology (nRFD = 0) and different branch lengths (KF > 0) (yellow), while the remaining 17,165 / 19,414 (88.5%) yielded the same topology (nRFD

= 0) and identical branch lengths (KF = 0) (green) (see panel a in figure below). In addition, we found that differences in branch lengths between trees with the same topology were much smaller than those observed between trees that differed in their topologies (see panel b in figure below).

These new results are included in the second paragraph of the section “The ML phylogenies of a considerable number of genes in phylogenomic datasets are irreproducible” in the revised manuscript.

If IQtree and Raxml both produce reproducible trees (i.e. their run 1 and run 2 are the same) do IQtree and Raxml always agree with each other?

AUTHORS' RESPONSE: We thank the reviewer for the question. To examine whether gene trees that are topologically reproducible by IQ-TREE are topologically identical to gene trees that are topologically reproducible by RAxML-NG, we compared IQ-TREE-inferred gene trees with RAxML-NG-inferred gene trees. We found that only 3,940 / 19,414 (20.3%) gene alignments yielded topologically identical phylogenies in Run1 and Run2 of IQ-TREE and in Run1 and Run2 of RAxML-NG (see figure below). These new results are included in the third paragraph of the section “The ML phylogenies of a considerable number of genes in phylogenomic datasets are irreproducible” in the revised manuscript.

If the processor and number of threads used were identical did this completely remove the issue of irreproducibility, or are there other factors at play as well? This is suggested but not explicitly stated.

AUTHORS' RESPONSE: Other factors are at play. Phylogenetic studies often report alignment, program, substitution model, and number of tree searches, but seldom report starting seed number, number of threads, and processor type. To better understand how these three typically unreported parameters (starting seed number, number of threads, and processor type) affect the irreproducibility of phylogenetic trees, we used three large representative studies to explicitly illustrate how each unreported parameter influences the reproducibility of inference.

As shown in the figure below (Figure 7 in the revised manuscript), we found that:

- a) If alignment, program, substitution model, and number of tree searches used to infer phylogenetic trees (which are typically reported) are unavailable, reproducibility is impossible.
- b) As long as alignment, program, substitution model, number of tree searches (all typically reported), and starting seed number (typically unreported) are available, the percentage of topologically reproducible genes can increase up to around 34% and 87% in IQ-TREE and RAxML-NG analysis, respectively.
- c) If all five parameters in (b), the number of threads, and the processor type used are all available, then all gene trees inferred are topologically reproducible in RAxML-NG (irrespective of the number of threads) and in IQ-TREE (for up to two threads). However, when using 3 or more threads in IQ-TREE, ~50% of genes remain irreproducible.

We have revised the “Discussion” section, including Figure 7, to make this point clear.

Figure 3 panel d suggests that processor differences explain all of the instances of irreproducibility, how does this reconcile with the finding that threads matters for iqtree? Or does Figure 3 only relate to the runs with 2 threads? If so, it may be good to put this in the caption.

AUTHORS' RESPONSE: Yes, the data used to generate Figure 3 concern runs with 2 threads. We revised the legend of Figure 3 to state this.

line 81

what is meant by "tree search number"? Is this the number of random starts?

AUTHORS' RESPONSE: Yes, "tree search number" means "the number of independent tree searches". We have fixed this in the revised manuscript.

line 92

what does it mean for two trees to be significantly different? I.e. what is the null hypothesis that is rejected by the AU test?

AUTHORS' RESPONSE: We thank the reviewer for the question. We used the AU test to test whether two topologically different trees inferred by Run1 and Run2 can equally explain the gene alignment (D); the null hypothesis H0 is that they can, and the alternative hypothesis (H1) is that they cannot (i.e., the Run1 and Run2 topologies are significantly different). If the p-value $P(D | H0)$ is smaller than 0.05, we reject H0 in favor of H1. We have revised this sentence to clarify this.

Line 108

what is normalised RF distance normalised by? Is a normalised RF of 0.58 saying that 58% of the splits differ?

AUTHORS' RESPONSE: Yes, a normalized RF value of 0.58 means that 58% of bipartitions differ between Run1 and Run2 trees. More generally, the normalized RF distance (nRFD) is the fraction of bipartitions (or internal branches) that differ between Run1 and Run2 trees. We have clarified this in the revised manuscript,

line 241-244

I don't understand the explanation given here

"This is likely because different orders of addition of the per-site log likelihoods when using three or more threads in IQ-TREE could result in two different phylogenies with different log-likelihood values."

addition is commutative so how can the result depend on the order?

AUTHORS' RESPONSE: Yes, addition is commutative. As you know, log likelihood values are very small and programs have to use the numerical precision. Let's say we have three threads and numerical precision is 0.0001. Log Likelihood for thread #1 is 0.01222; Log Likelihood for thread #2 is 0.01223; Log Likelihood for thread #3 is 0.01228.

a) If we add threads in the order #1, #2, #3, the commutative likelihood value is $(0.01222 + 0.01223 = 0.0244) + 0.01228 = 0.0366$.

b) If we add threads in the order #1, #3, #2, the commutative likelihood value is $(0.01222 + 0.01228 = 0.0245) + 0.01223 = 0.0367$.

We revised the sentence to clarify this. It now reads: "This is likely because different orders of commutative addition of the per-site log likelihoods when using three or more threads in IQ-TREE could result in two different phylogenies with different log-likelihood values".

line 310

searchers -> searches

AUTHORS' RESPONSE: We changed "searchers" to "searches".

line 370

what is a kernel vector?

AUTHORS' RESPONSE: We thank the reviewer for the question. We changed “kernel vector” to “kernel instruction”.

line 593

produces -> produce

AUTHORS' RESPONSE: We changed “produces” to “produce”.

Reviewer #2 (Remarks to the Author):

The submission addresses the issue of reproducibility of phylogenetic tree inference. The authors consider several large published eukaryotic datasets. On each set of related genes, they infer two trees using the same program and parameters and check to which extent the topology between them agree. They observe differences in a substantial number of cases, and conclude that phylogenetic inference suffers from widespread irreproducibility, and conclude that authors should disclose random seed number, number of treads, and processor type used.

The work addresses an important problem, and the results obtained are thought-provoking.

I have one main concern, described below in detail, which I think could be satisfactorily addressed in a revision of the manuscript with more discussion and a restatement of the main conclusion.

Finally, note that I have reviewed an earlier version of the manuscript. I thank the authors for addressing some of the minor points I raised in my previous report. The points included in this report are, as far as I can tell, yet to be addressed.

AUTHORS' RESPONSE: We thank the reviewer for their comments on two different versions of this study, for appreciating the importance of our work, and for the constructive feedback.

Main point: what degree of reproducibility is desirable? and implications of the study

Phylogenetic methods are very broadly used, but the issue of interpreting trees, and in particular gauging the uncertainty around the inferred topology, is challenging even for experts (see e.g. intense debates around the placement of Ctenophora). Phylogenomic resources are complex pipelines, of which each step is afflicted by random and systematic errors.

The issue of reproducibility is thus highly relevant. But what exactly should be reproducible? In this paper, the authors have taken the view that reproducibility means to obtain the same tree topology every time. But, as I just mentioned, trees are subject to inference uncertainty, which is typically gauged using branch support measures such as bootstrap support. If the difference between two topologies is in uncertain parts of the tree (i.e. within the confidence interval, so to speak), is this still a reproducibility issue?

AUTHORS' RESPONSE: We fully agree with the reviewer's view that irreproducibility in phylogenetic inference can be examined not just for the tree's topology, but also other aspects, such as the tree's branch lengths, and the tree's branch support values (e.g., bootstrap support values).

Given this broader view of the reproducibility of phylogenetic inference, the reviewer asks whether the differences in topology are localized in parts of the tree that are weakly supported. To examine whether the bipartitions that differ between Run1 and Run2 trees were highly supported, we evaluated the reliability of each internal branch using 1,000 ultrafast bootstrap replicates (the option “-bb 1000”) and 100 standard bootstrap replicates (the option “--bs-trees 100”) for IQ-TREE and RAxML-NG, respectively. Since examining all 19,414 gene alignments from 15 phylogenomic datasets was computationally intractable, we sampled the first 100 genes from each dataset. For each gene alignment, we inferred gene trees twice with identical settings.

As shown in the figure below (Supplementary Figure 9 in the revised manuscript), we found that:

- a) The genes that yield topologically irreproducible phylogenies have lower bootstrap support values than genes that yield topologically reproducible phylogenies (see panel a in figure below).
- b) By comparing the branch support of conflicting internal branches against those of congruent internal branches within irreproducible genes (i.e., genes that yield two topologically different trees in Run1 and Run2), we found that conflicting internal branches received lower bootstrap support values than congruent internal branches (see panel b in figure below).
- c) Furthermore, we found that the percentage of conflicting internal branches with strong bootstrap support (rapid bootstrap value ≥ 95 in IQ-TREE analysis, bootstrap value ≥ 70 in RAxML-NG analysis), is much lower than that of congruent internal branches (see panel c in figure below).

These new results were included in the second paragraph of the section “Low phylogenetic informativeness, multithreading, and processor type contribute to irreproducibility” in the revised manuscript.

Given that these results suggest that reproducibility disproportionately affects parts of the topology that are weakly supported, the reviewer asks: “If the difference between two topologies is in uncertain parts of the tree (i.e. within the confidence interval, so to speak), is this still a reproducibility issue?”

We believe these results do indeed raise a reproducibility issue for the following reason. If we use the Run1 and Run2 sets of individual gene trees (generated using identical parameter settings in either IQ-TREE or RAxML-NG), we find that 9 / 15 (60%) phylogenomic datasets produced topologically different coalescent-based ASTRAL species phylogenies (see new Table 1 in our revised manuscript). Notably, several of these datasets produced topologically different coalescent-based ASTRAL species phylogenies even if branches below 50% bootstrap support were collapsed in Run1 and Run2 gene trees (see new Table 2 in our revised manuscript). Thus, the observed irreproducibility is sufficiently strong to alter the results of state-of-the-art phylogenomic approaches, even when sources of uncertainty are removed.

Because of the vast number of possible tree topologies (for a set of just 20 taxa, there are 221,643,095,476,699,771,875 possible unrooted binary topologies) and the complexity of optimising over the often rugged likelihood "landscape" of the topological space, it can be very difficult to find the global optimum. Inferred trees should never be considered at face value, but rather interpreted in light of the inference uncertainty.

AUTHORS' RESPONSE: We agree that there is uncertainty in inference that stems from the fact that we are performing heuristic searches. However, we think that our results on irreproducibility reveal uncertainty in inference that is well over and above that stemming from the fact that we are conducting heuristic searches.

Briefly, the widely used default values for log likelihood epsilon and number of different starting trees in IQ-TREE and RAxML-NG heuristic searches are 0.1 and 1, and 0.1 and 20, respectively. We instead used a log likelihood epsilon value of 0.0001 and 20 different starting trees to make our searches more precise and reduce the uncertainty introduced by heuristic search. To examine whether the degree of irreproducibility of phylogenetic trees was influenced by the values of parameters that influence heuristic search precision, we examined the impact of different numbers of starting trees (20, 50, and 100) on reproducibility of gene phylogenies for three representative phylogenomic studies.

As shown in the figure below (Figure 4 in the revised manuscript), we found that numbers of tree searches have generally little effect on irreproducibility of single-gene phylogenetic trees in both programs. These new results were included in the last paragraph of the section "The ML phylogenies of a considerable number of genes in phylogenomic datasets are irreproducible" in the revised manuscript.

Thus, unless the purpose is to debug code, I see relatively limited value in enforcing the ability to reproduce specific rounding-off errors or optimisation paths, as long as these

differences don't materially affect the output trees *after accounting for inference uncertainty*.

AUTHORS' RESPONSE: We thank the reviewer for the comment. To investigate the effect of irreproducibility of single gene trees on output trees, we used the Run1 and Run2 sets of individual gene trees (generated using identical parameter settings in either IQ-TREE or RAxML-NG) to infer coalescent-based ASTRAL species phylogenies. We find that 11 / 15 (73%) phylogenomic datasets produced topologically different coalescent-based ASTRAL species phylogenies. Thus, the observed irreproducibility is sufficiently strong to alter the output trees (see Table 1 in revised manuscript).

We next collapsed branches with low bootstrap support values ($\leq 50\%$) for Run1 and Run2 gene trees, and then used these partially multifurcating genes trees to infer coalescent-based ASTRAL trees. As shown in the Table below (Table 2 in the revised manuscript), we found that the percentage of conflicting bipartitions between coalescent-based ASTRAL species phylogenies from the Run1 and Run2 sets of individual gene trees, ranges from 0.29% to 18.18%, even though we accounted for the inference uncertainty thought collapsing branches with low bootstrap support values ($\leq 50\%$). Although collapsing branches with low bootstrap support values eliminated the irreproducibility of coalescent-based ASTRAL trees for three and four phylogenomic datasets in IQ-TEE and in RAxML-NG, respectively, eight phylogenomic datasets in IQ-TEE and seven phylogenomic datasets in RAxML-NG still yielded topologically different ASTRAL species phylogenies. Thus, the observed irreproducibility is sufficiently strong to alter the output trees even if we remove poorly supported branches from our gene trees.

Table 2 Comparisons of coalescent-based ASTRAL species phylogenies with and without collapsing poorly supported bipartitions in replicate gene trees.

Dataset	Program	Without collapsing branches		With collapsing branches	
		Conflicting bipartitions (%) ^a	Highly conflicting bipartitions (%) ^b	Conflicting bipartitions (%) ^a	Highly conflicting bipartitions (%)
Animal: Bees	IQ-TREE	22.46	1.87	18.18	1.60
Animal: Birds	IQ-TREE	6.09	0	6.60	0
Animal: Butterflies	IQ-TREE	7.35	0.49	5.39	0.74
Animal: Lizards	IQ-TREE	7.69	0	0	0
Animal: Marine fishes	IQ-TREE	7.69	0	13.68	0
Animal: Rodents	IQ-TREE	0	0	0	0
Plant: Cardueae	IQ-TREE	12.20	0	12.20	0
Plant: Caryophyllales	IQ-TREE	0	0	0	0
Plant: Green plants	IQ-TREE	5.70	0.38	5.53	0.55
Plant: Jaltomata	IQ-TREE	8.33	0	0	0

Plant: Protea	IQ-TREE	14.52	0	3.23	0
Fungi: Aspergillaceae	IQ-TREE	0	0	0	0
Fungi: Budding yeasts	IQ-TREE	0.29	0	0.29	0
Fungi: Hanseniaspora	IQ-TREE	0	0	0	0
Fungi: Rhizoplaca	IQ-TREE	3.57	0	0	0
Animal: Bees	RAxML-NG	27.27	1.87	8.02	0
Animal: Birds	RAxML-NG	4.06	0	0	0
Animal: Butterflies	RAxML-NG	6.86	0.49	6.86	0
Animal: Lizards	RAxML-NG	19.23	0	0	0
Animal: Marine fishes	RAxML-NG	3.42	0	5.98	0
Animal: Rodents	RAxML-NG	0	0	0	0
Plant: Cardueae	RAxML-NG	12.20	0	3.66	0
Plant: Caryophyllales	RAxML-NG	0	0	0	0
Plant: Green plants	RAxML-NG	0.26	0	1.02	0.09
Plant: Jaltomata	RAxML-NG	8.33	0	0	0
Plant: Protea	RAxML-NG	14.52	0	8.06	0
Fungi: Aspergillaceae	RAxML-NG	0	0	0	0
Fungi: Budding yeasts	RAxML-NG	0.29	0	0.59	0
Fungi: Hanseniaspora	RAxML-NG	0	0	0	0
Fungi: Rhizoplaca	RAxML-NG	3.57	0	0	0

For each dataset, coalescent-based ASTRAL trees were reconstructed from the Run1 and Run2 sets of individual gene trees without and with collapsing branches with low bootstrap support values ($\leq 50\%$); both Run1 and Run2 used identical settings, including substitution model, random starting seed number, number of threads of execution, number of independent tree searches, number of bootstrap replicates, and ML program. ^a Percentage of conflicting bipartitions between coalescent-based ASTRAL species phylogenies inferred using Run1 and Run2 gene trees. ^b Percentage of highly conflicting bipartitions (LPP $\geq 90\%$) between coalescent-based ASTRAL species phylogenies inferred using Run1 and Run2 gene trees.

We have included these new results in the second paragraph of the section “Irreproducibility of coalescent- and concatenation-based phylogenomic inference” in the revised manuscript.

There is, however, a real cost in demanding such narrow-sense reproducibility. As the authors themselves observed, avoiding variation due to CPU-specific optimisations mean that programs will typically run more slowly. More importantly, it would impose strong constraints on computing on clusters, as this would require keeping track of the number of jobs, order of computations, and CPU types employed for each subtask.

AUTHORS’ RESPONSE: We agree with the reviewer that the cost is substantial. Given that phylogenomic analyses are computationally expensive (e.g., analysis of each of the 15 phylogenomic datasets took an average of ~3,400 CPU hours), it would be counterproductive to conduct phylogenomic inference on specific nodes / processors. Instead, we believe that authors should provide the log file of each analysis, which contains the values of key parameters (e.g., type of processor, number of threads, and

random seed) for reproducibility so that they can evaluate any differences in the results of analyses of the same dataset. This is straightforward to do and requires relatively low effort and cost. Of course, whether our suggestion is adopted by individual researchers and the phylogenetics community is for them to decide. However, we believe the levels of irreproducibility observed in our analyses justify raised this suggestion in the last sentence in the abstract and in the last paragraph of the “Discussion”.

So while I think that it is interesting to demonstrate that multiple runs of the same data, using the same parameters, can result in different point estimates of the topology (which the authors convincingly show), I feel the conclusion should not be that all authors now need to systematically report their choice of seed and CPU types, and deactivate multithreading, but rather that more work is needed to assess whether these differences are significant in light of the uncertainty, and whether they might affect conclusions drawn from the data.

AUTHORS’ RESPONSE: We thank the reviewer for the suggestion. We have already described in our previous responses that irreproducibility can yield different output trees in phylogenomic analyses. To further examine whether the observed differences are significant, we used the AU test to see whether two topologically different trees inferred two replicates (Run1 and Run2) can equally explain (the null hypothesis H_0) the gene alignment (D). If the p-value $P(D | H_0)$ is smaller than equal to 0.05, we reject the null hypothesis H_0 , suggesting the Run1 and Run2 gene trees are significantly different.

As shown in the figure below (Supplementary Figure 4 in the revised manuscript), we found that the Run1 and Run2 topologies for 302 / 3,515 (8.59%) irreproducible single-gene ML phylogenies generated by IQ-TREE and 457 / 1,813 (25.2%) irreproducible phylogenies generated by RAxML-NG were significantly different (the approximately unbiased (AU) test; P-value ≤ 0.05). We believe that these results, together with the results of the new Table 1 (that shows that this irreproducibility can affect coalescent-based species inference), suggest that they can affect the conclusions drawn from the data. Clearly more work is necessary (and we have now stated this in the penultimate paragraph of the “Discussion” section, but we believe that our recommendation to provide the log file of each analysis is both reasonable (given our results) and involves low effort and cost.

We have included these results in the fourth paragraph of the section “The ML phylogenies of a considerable number of genes in phylogenomic datasets are irreproducible” in the revised manuscript.

It might be possible to start answering this question from the results at hand. For instance, looking at the supplementary materials, I could see that some of the differences in the ASTRAL trees are on nodes with high branch support, which I think is noteworthy and cause for concern. See also the discretionary point below.

AUTHORS' RESPONSE: We thank the reviewer for the suggestion. In addition to the irreproducibility of single-gene trees, we included an additional section entitled “Irreproducibility of coalescent- and concatenation-based phylogenomic inference” and a new table (Table 1) to present the irreproducibility of coalescent-based ASTRAL trees and concatenation-based ML trees inferred with identical parameter settings. Since the single-gene trees that are irreproducible across two replicates (Run1 and Run2) are the input data for inferring coalescent-based ASTRAL trees, we also examined whether accounting for the inference uncertainties (e.g., collapsing branches with low bootstrap values) of single-gene tree eliminated the topological differences between two coalescent-based ASTRAL species trees inferred from Run1 and Run2 gene trees (Table 2). We found that this practice reduced, but did not eliminate, differences in the coalescent-based ASTRAL species trees. Thus, we agree with the reviewer that these results are both “noteworthy and cause for concern”.

Minor points

* The result that RAxML-NG gives consistent results under multithreading but not in multiprocessing mode is intriguing. It may be worth inspecting the code or contacting the authors to find out why.

AUTHORS' RESPONSE: We thank the reviewer for the suggestion. After checking the RAXML-NG's wiki on the GitHub (<https://github.com/amkozlov/raxml-ng/wiki/Parallelization>), we found that the distinct vectorized kernels (i.e., multiprocessing modes) might lead to slightly different likelihood values due to numerical precision, therefore inferred gene trees could be topologically different (please see their vector instructions on the GitHub repository).

* Just to illustrate the practical difficulty of implementing the authors' own recommendation, I was unable to find the random seed values used to reconstruct the simulation in the "3_simulation_on_CHTC" supplementary file on FigShare. (I feel bad nitpicking this, given that I feel the authors already went to considerable length to provide all supporting data, and this is a very minor oversight.)

AUTHORS' RESPONSE: We thank the reviewer for pointing this out. Considering the practical difficulty of providing the values of typically unreported parameters (e.g., random starting seed values, number of threads, processor), we have revised our text and now suggest (rather than recommend) that authors provide the log file of each analysis, which contain the values of all these key parameters. Implementation of this suggestion should be easy and would provide then necessary information to other researchers interested in replicating a study's findings. We have discussed this in the last paragraph of the "Discussion".

In the initial version, the random seed values used to simulate gene alignments were given in the file "Seq_Gen_run.bat" that was in the compressed zip file "guide_trees_alignments.tar.gz". The random seed values used to infer single gene trees were given in the information tables in the compressed zip file "summary_data.tar.gz". In the revised manuscript, we moved "Seq_Gen_run.bat" file and the information table files to the parent "3_simulation_on_CHTC" folder.

* discretionary point: It would be interesting to quantify whether how that the difference in architectures and number of processors causes a difference larger than different random seeds

AUTHORS' RESPONSE: We thank the reviewer for the suggestion. To answer this question, we included two additional analyses (see the new scenarios III and IV in the figure below, which corresponds to Fig. 7 in the revised manuscript). In scenario III, all parameters are identical between the two replicates (Run1 and Run2), but random seed number is not. In scenario IV, all parameters, including starting seed number, are identical between the two replicates except number of threads and

processor. Since running all 19,414 gene alignments from 15 phylogenomic datasets was computationally intractable, we sampled the first 200 genes from each of three large representative studies in animals, plants, and fungi for IQ-TREE (in yellow) and RAxML-NG (in blue), respectively.

Our results show that random seed number differences lead to much lower reproducibility than difference in architectures and number of processors. Thus, random seed number has greater influence on reproducibility than architecture and number of processors. These new results have been included in fifth paragraph of the “Discussion” in the revised manuscript.

Christophe Dessimoz

Reviewer #3 (Remarks to the Author):

The paper looks at how reproducible two phylogenetic analysis packages are, in terms of producing the same outcome when starting conditions are identical. It uses real and simulated data to demonstrate that the packages are not as repeatable as one would hope.

The programs the authors studied are primarily designed for building phylogenies quickly, the philosophy driving these programs is speed over accuracy, they are known to produce highly unreliable results, the program may find the highest likelihood less than 50% of the time for a given data set, as the authors of this study previously pointed out in 2018.

AUTHORS' RESPONSE: We respectfully disagree with the reviewer's assessment for several reasons.

First, the purpose of our study was to evaluate whether two independent runs (Run1 and Run2) with *exactly the same parameters* (including ML program, starting tree, tree rearrangement strategy, substitution model, number of CPU cores etc.) generate identical results. We do not believe that any other study has addressed this question on irreproducibility or reported our findings.

Second, our use of two independent runs, Run1 and Run2, with *exactly the same parameters* means that the Run1 and Run2 trees should yield the same tree (irrespective of whether that tree is accurate); to put it in a different way, our experiments are examining the precision or reproducibility of phylogenetic inference and not its accuracy *per se* (although we recognize that these two concepts are connected).

Third, ranking programs based on how often they achieve the highest likelihood score (Figure 2 in Zhou et al. 2018, Mol Biol Evol), the two programs used in this study (IQ-TREE and RAxML) have the best performance of finding the best-scoring trees compared to other ML programs, such as PhyML and FastTree.

Fourth, the overall mean probability of obtaining the tree with the highest likelihood score is greater than 75% in both programs when the number of tree searches is 10 in IQ-Tree and RAxML. Our study used an even more intensive tree search strategy (20 tree searches).

Finally, to examine whether the degree of irreproducibility of phylogenetic trees would decrease as the accuracy of tree search increased, we examined the impact of using different numbers of tree searches (20, 50, and 100) on the reproducibility of gene phylogenies for three representative phylogenomic studies. As shown in the figure below (Figure 4 in the revised manuscript), we found that numbers of tree searches have generally little effect on irreproducibility of single-gene phylogenetic trees in both programs.

The title of the paper “Irreproducibility in molecular phylogenetics” is not justified given the analysis. The authors study two programs out of the hundreds available, the two they chose were known to be inaccurate. The authors then link this to the reproducibility crisis. They note that “... irreproducibility can be completely eliminated by turning off the kernel auto-detection option in RAxML-NG ...” but at the cost of a 2.4 speed penalty. If a crisis can be avoided by setting an option, it probably is not a crisis.

AUTHORS’ RESPONSE: We thanks the reviewer for these comments.

The first comment concerns our title. We changed “Irreproducibility in molecular phylogenetics” to “Irreproducibility of maximum likelihood phylogenetic inference”, which we agree is more accurate.

The second comment is that the two maximum likelihood (ML) programs are inaccurate. We have already explained in response to the reviewer’s previous comment why we disagree. We further note that it is generally well accepted that maximum likelihood methods are more reliable / accurate than distance and parsimony methods (e.g., see Yang and Rannala 2012, Nat Rev Genet). It is also the case that maximum likelihood methods are widely used in phylogenomic studies. The papers that introduced IQ-TREE v1, RAxML v8, PhyML v3, and FastTree v2 were collectively cited a total of 7,510 times in the past nine months according to Google Scholar. Among these four programs, IQ-TREE and RAxML have the best performance of finding the best-scoring trees (please see the Figure 2 from Zhou et al. 2018, Mol Biol Evol). Thus, we believe that gene trees inferred by IQ-TREE and RAxML are not only very commonly used but are also considered accurate; therefore, evaluating their irreproducibility is a very interesting question with potentially major implications for maximum likelihood phylogenetic inference.

The third comment argues that the observed irreproducibility is not of concern. We respectfully disagree. The current version of IQ-TREE does not have an option to turn off the autodetect of CPU optimization; RAxML-NG does offer the option, but this

comes at the cost of a 2.4 speed penalty, which is substantial for phylogenomic studies. As the sizes of phylogenomic datasets continue to increase, it will be essential to maximize the capabilities of CPU processors, whose architecture and number of threads are variable across supercomputer clusters. We also note that we are the first to report that the observed irreproducibility also affects coalescent-based ASTRAL species tree inference, which is one of the two major approaches (together with coalescence) for species tree inference in phylogenomics.

The authors suggest that 100% reproducibility can be obtained using recipe 10 in figures 6. It does not take into account a number of factors that will influence the results, especially if low level CPU instructions such as single instruction multiple data (SIMD) are part of the problem, to make the results reproducible, large amounts of technical information (which were not supplied) will be needed, BIOS versions, microarchitecture version, operating system / kernel version, compiler manufacturer and version, what version of the libraries the programs are using (this will be a long list with lots of dependencies), what implementation and version of MPI, compiler build options, etc. If reproducibility is the goal, technologies such as containers / VM's will be needed.

AUTHORS' RESPONSE: We agree that different containers might influence the compilation of program, thence the output could be different. We have now added a discussion of this point in the third paragraph of the "Discussion" section in the revised manuscript.

This paper is a technical note on the quirks of these two programs, not a look at reproducibility in molecular phylogenetics.

AUTHORS' RESPONSE: We respectfully disagree. We found this important phenomenon of irreproducibility of both gene trees and species trees in maximum likelihood inference, which is one of the major statistical methods in molecular phylogenetics and phylogenomics, explored underlying causes with empirical and simulated data, and listed suggestions how to improve the reproducibility of gene trees. We believe this work will be of substantial interest to phylogeneticists and evolutionary biologists in general.

Reviewers' Comments:

Reviewer #1:

Remarks to the Author:

I thank the authors for their very comprehensive response to my comments on the earlier manuscript. I think the new analyses that have been performed will broaden the interest in these results. In particular, it was very interesting (and somewhat alarming) to see that suppressing branches with low bootstrap support did not entirely alleviate the issue of irreproducibility.

I think this is an important study that should attract wide interest in the community of developers and users of phylogenetic methods.

The paper is very clearly written and the key points are convincingly demonstrated. I have no further suggestions.

Barbara Holland

Reviewer #2:

Remarks to the Author:

I thank the authors for thoroughly addressing all my comments. I found the additional analyses informative, and the answers to my questions satisfactory. Thanks also for formulating more practical recommendations. I have no further reservation and am thus happy to endorse this revised manuscript.

Christophe Dessimoz